# DiffMoE: Dynamic Token Selection for Scalable Diffusion Transformers

## Abstract

Diffusion Transformers (DiTs) have become the leading architecture for visual generation tasks. However, their uniform treatment of inputs across different conditions and noise levels overlooks the inherent heterogeneity of the diffusion process. While recent mixture-of-experts (MoE) in diffusion approaches attempt to address this limitation, they struggle to achieve significant improvements due to their restricted token accessibility and fixed computational patterns. We present **DiffMoE**, a novel MoE-based architecture that enables experts to access global token distributions through a **batch-level global token pool** during training, promoting specialized expert behavior. To unleash the full potential of inherent heterogeneity, DiffMoE incorporates **capacity predictor** and **dynamic threshold** that adpatively allocates computational resources based on noise levels and sample complexity. Through comprehensive evaluation, DiffMoE achieves state-of-the-art performance among diffusion transformers on ImageNet benchmark, substantially outperforming both dense architectures with $3\times$ activated parameters and existing MoE approaches while maintaining $1\times$ activated parameters. Our approach demonstrates efficacy not only in class-conditional generation but also in more complex tasks such as text-to-image synthesis, outperforming both dense models and various DiT baselines. This underscores its broad applicability across diverse diffusion model applications.

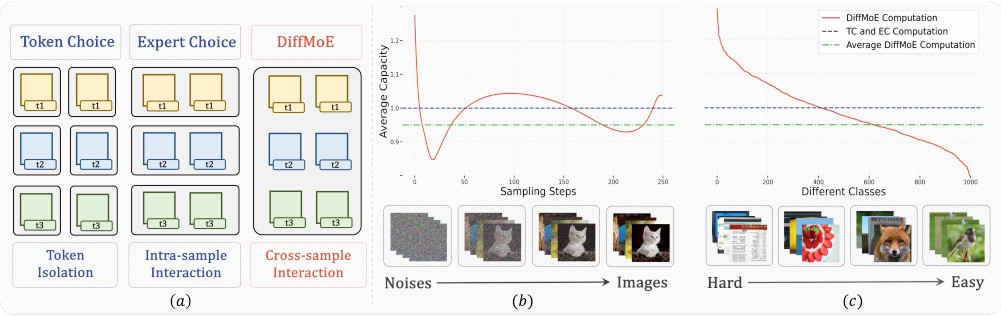

Figure 1: **Token Accessibility and Dynamic Computation.** **(a)** Token accessibility levels from token isolation to cross-sample interaction. **(b)** Computational dynamics during diffusion sampling, showing adaptive computation from noise to image. **(c)** Class-wise computation allocation from hard (technical diagrams) to easy (natural photos) tasks. Results from DiffMoE-L-E16-Flow (700K).

## 1 Introduction

The Mixture-of-Experts (MoE) framework (Lepikhin et al., 2020; Shazeer et al., 2017) has emerged as a powerful paradigm for enhancing overall multi-task performance while maintaining computational efficiency. This is achieved by combining multiple expert networks, each focusing on a distinct task, with their outputs integrated through a gating mechanism. In language modeling, MoE has achieved performance comparable to dense models of $2\times - 3\times$ activated parameters (DeepSeek-AI et al., 2024; MiniMax et al., 2025; Muennighoff et al., 2024). The current MoE primarily follows two gating paradigms: Token-Choice (TC), where each token independently selects a subset of experts for

processing; and Expert-Choice (EC), where each expert selects a subset of tokens from the sequence for processing.

Diffusion (Ho et al., 2020; Rombach et al., 2022; Podell et al., 2023; Song et al., 2021) and flow-based (Ma et al., 2024; Esser et al., 2024b; Liu et al., 2023) models inherently represent multi-task learning frameworks, as they process varying token distributions across different noise levels and conditional inputs. While this heterogeneity characteristic naturally aligns with the MoE framework's ability for multi-task handling, existing attempts (Fei et al., 2024; Sun et al., 2024; Yatharth Gupta, 2024; Sehwag et al., 2024) to integrate MoE with diffusion models have yielded suboptimal results, failing to achieve the remarkable improvements observed in language models. Specifically, Token-choice MoE (TC-MoE) (Fei et al., 2024) often underperforms compared to conventional dense architectures under the same number of activations; Expert-choice MoE (EC-MoE) (Sehwag et al., 2024; Sun et al., 2024) shows marginal improvements over dense models, but only when trained for much longer.

We are curious about what fundamentally limits MoE's effectiveness in diffusion models. Our key finding reveals that **global token distribution accessibility is crucial for MoE success in diffusion models, necessitating the model learn and dynamically process the tokens from different noise levels and conditions**, as illustrated in Figure 1(b)(c). Previous approaches have neglected this crucial component, resulting in compromised performance. Specifically, Dense models and TC-MoE isolates tokens, preventing them from interacting with others during expert selection, while EC-DiT restricts intra-sample token interaction, which fails to access other samples with different noise levels and conditions. These limitations hinder the model's ability to capture the full spectrum of the heterogeneity inherent in diffusion processes.

To address these limitations, we introduce **DiffMoE**, a novel architecture that features a **batch-level global token pool** for enhanced **cross-sample token interaction** during training, as illustrated in Figure 2. This approach approximates the complete token distribution across different noise levels and samples, facilitating more specialized expert learning through comprehensive global token information access. Our empirical analysis demonstrates that the global token pool is instrumental in accelerating loss convergence. Compared to dense models with an equivalent number of activation parameters, it achieves significantly faster convergence.

However, conventional MoE inference strategies, which maintain fixed computational resource allocation across different noise levels and conditions, fail to fully leverage the potential of DiffMoE's batch-level global token pool. To optimize token selection during inference, we propose a **capacity predictor** that dynamically adjusts resource allocation. This adaptive mechanism learns from training-time token routing patterns, efficiently distributing computational resources between complex and simple cases. Furthermore, we implement a **dynamic threshold** at inference time to achieve flexible performance-computation trade-offs. By integrating the global token pool and capacity predictor, **DiffMoE achieves superior performance over dense models with** $3\times$ **activated parameters** while maintaining efficient scaling properties (See Table 2). Our approach offers extra several advantages over existing methods: it eliminates the potentially detrimental load balancing losses present in TC-MoE and overcomes the intra-sample token selection constraints of EC-MoE, resulting in enhanced flexibility and scalability. Extensive empirical evaluations demonstrate DiffMoE's superior scaling efficiency and performance improvements across diverse diffusion applications.

Our contributions can be summarized as follows:

(1) We identify the critical role of global token distribution accessibility in enabling dynamic token selection for MoE-based diffusion models.

(2) We propose DiffMoE, a scalable framework with a global token pool, capacity predictor, and dynamic threshold for efficient computation.

(3) We demonstrate superior performance on ImageNet and T2I benchmarks via dynamic computation allocation without sacrificing efficiency.

(4) We provide extensive experiments across diverse diffusion tasks, validating the effectiveness and generality of our approach.

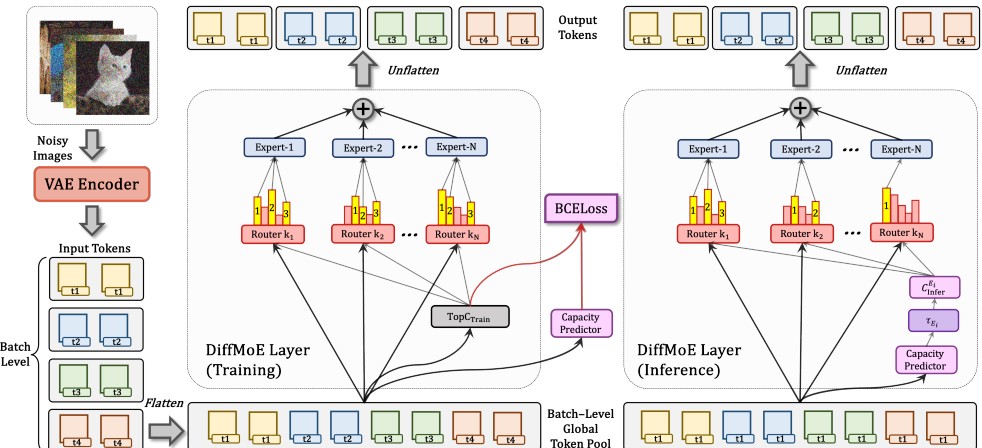

Figure 2: **DiffMoE Architecture Overview.** DiffMoE flattens tokens into a batch-level global token pool, where each expert maintains a fixed training capacity of $C_{\text{train}} = 1$. During inference, a dynamic capacity predictor adaptively routes tokens across different sampling steps and conditions. Different colors denote tokens from distinct samples, while $t_i$ represents corresponding noise levels.

## 2 METHODOLOGY

**Diffusion Models.** Diffusion models (Ho et al., 2020; Rombach et al., 2022; Sohl-Dickstein et al., 2015; Song et al., 2021) are a powerful family of generative models, which can transform the noise distribution $p_1(\mathbf{x})$ to the data distribution $p_0(\mathbf{x})$. The diffusion process can be represented as: $\mathbf{x}_t = \alpha_t \mathbf{x}_0 + \sigma_t \epsilon$, $t \in [0,1]$, $\epsilon \sim \mathcal{N}(0, \mathbf{I})$, Where $\alpha_t$ and $\sigma_t$ are monotonically decreasing and increasing functions of $t$, respectively. The marginal distribution $p_1(\mathbf{x})$ converges to $\mathcal{N}(0, \mathbf{I})$, when $\alpha_1 = \sigma_0 = 0, \alpha_0 = \sigma_1 = 1$.

To train a diffusion model, we can use the denoising score matching method (Song et al., 2021) which constructs a score prediction model $\epsilon_\theta(\mathbf{x}_t, t)$ to estimate the scaled score function $-\sigma_t \nabla_{\mathbf{x}} \log p_t(\mathbf{x}_t)$ with training objective formulated in Eq. 21. Sampling from a diffusion model can be achieved by solving the reverse-time SDE or the corresponding diffusion ODE (Song et al., 2021) in an iterative manner. Recently, flow-based models (Liu et al., 2022a; Lipman et al., 2022; Esser et al., 2024a) have shown superior performance through alternative training objective formulated in Eq. 30 while maintaining the same architecture as DiT (Peebles & Xie, 2023a). Sampling from a flow-based model can be achieved by solving the probability flow ODE.

**Mixture of Experts.** Mixture of Experts (MoE) (Shazeer et al., 2017; Cai et al., 2024) is based on a fundamental insight: different parts of a model can specialize in handling distinct tasks. By selectively activating only relevant components, MoE enables efficient scaling of model capacity while maintaining computational efficiency. MoE layers generally consist of $N$ experts, each implemented as a Feed-Forward Network (FFN) with identical architecture, denoted by $E_1(\mathbf{x}), \ldots, E_N(\mathbf{x})$ with input $\mathbf{x}$. A routing matrix $\mathbf{W}_r \in \mathbb{R}^{D \times N}$ is used to calculate token-expert affinity matrix:

$$\mathbf{M} = \texttt{softmax}_E(\mathbf{x}\mathbf{W}_r), \quad \mathbf{x} \in \mathbb{R}^{B \times S \times D}, \tag{1}$$

where $B$ is the batch size, $S$ is the token length of one sample, $D$ is the hidden dimension, $\texttt{softmax}_E$ denotes the $\texttt{softmax}$ operation along the expert axis. There are two common gating paradigms: Token-Choice (TC) (Shazeer et al., 2017; Fei et al., 2024) and Expert-Choice (EC) (Zhou et al., 2022; Sun et al., 2024). For TC, each token of each sample individually selects $\texttt{top-}K$ experts via a gating function, the gating function and output of TC-MoE layers are defined as follows:

$$\mathbf{G}_{s,i}^{TC} = \begin{cases} \mathbf{M}_{s,i}, & \mathbf{M}_{s,i} \in \texttt{top-}K(\{\mathbf{M}_{s,i}\}_{i=1}^N) \\ 0, & \text{otherwise} \end{cases} \tag{2}$$

$$\mathbf{y}_s = \sum_{i=1}^{K} \mathbf{G}_{s,i}^{TC} E(\mathbf{x}_s), \mathbf{x}_s \in \mathbb{R}^{1 \times D}, s \in \{1, \ldots, S\}. \tag{3}$$

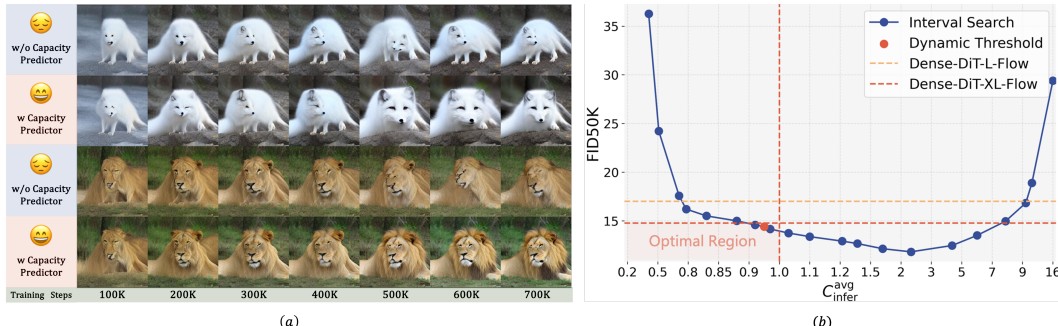

Figure 3: **(a) Effectiveness of Capacity Predictor.** Comparison of sampling strategies with DiffMoE-L-E16-Flow (Batch Size = 1). For each group, **Top:** Sampling w/o Capacity Predictor (with Fixed TopK Method.) **Bottom:** Sampling with Capacity Predictor. **(b) Different Threshold Methods:** We employ two distinct approaches for threshold determination: dynamic threshold (red point) and interval search (blue points). Visualization using DiffMoE-L-E16-Flow (700K).

Different from TC, EC makes every expert selects $K'$ tokens from each sample of the input $\mathbf{x} \in \mathbb{R}^{B \times S \times D}$. The gating function $G_{s,i}^{EC}$ can be implemented analogously to Eq. 2, with the modification that the $\texttt{top}$ operation selects $K'$ tokens along the token dimension, i.e. $S$. Similar to Eq. 3, the output for a token $\mathbf{x}_s$ of EC-MoE layers can be calculated as: $\mathbf{y}_s = \sum_{i=1}^{N} \mathbf{G}_{s,i}^{EC} E(\mathbf{x}_s), \mathbf{x}_s \in \mathbb{R}^{1 \times D}$.

Both TC and EC struggle to achieve significant improvements comparing with dense models due to their restricted token accessibility and fixed computational patterns.

## 2.1 DiffMoE: Dynamic Token Selection

**Batch-level Global Token Pool.** Since MoE architectures replace FFN layers, both TC and EC paradigms in diffusion models are inherently limited to processing tokens within individual samples, where gating mechanisms operate exclusively on tokens sharing identical conditions and noise levels. This architectural constraint not only prevents experts from learning crucial contrastive patterns but, more fundamentally, restricts their access to the global token distribution that characterizes the full spectrum of the diffusion process. To capture this essential global context, we introduce a *Batch-level Global Token Pool* for DiffMoE by flattening batch and token dimensions, enabling experts to access a comprehensive token distribution spanning different noise levels and conditions. This design, which simulate the true token distribution of the entire dataset during training, can be formulated as follows:

$$\mathbf{x} \in \mathbb{R}^{B \times S \times D} \rightarrow \mathbf{x}_{\text{pool}} \in \mathbb{R}^{BS \times D}. \tag{4}$$

During the training phase, we push expert $E$ to select $K_{\text{train}}^E$ tokens, forcing each expert to capture the characteristics of tokens from different conditional information and noise levels, while keeping expert load balance during training. The corresponding batch-level global token-expert affinity matrix will be calculated as follows:

$$\mathbf{M}^{Dy} = \mathbf{x}_{\text{pool}} \mathbf{W}_r, \mathbf{x}_{\text{pool}} \in \mathbb{R}^{BS \times D}, \mathbf{W}_r \in \mathbb{R}^{D \times N}. \tag{5}$$

Then, using $\mathbf{M}^{Dy} \in \mathbb{R}^{BS \times N}$, the gating value of MoE and the output of DiffMoE layers can be computed as follows:

$$\mathbf{G}_{s,i}^{Dy} = \begin{cases} \mathbf{M}_{s,i}^{Dy}, & \mathbf{M}_{s,i}^{Dy} \in \texttt{top-}K_{\text{train}}^E(\{\mathbf{M}_{s,i}^{Dy}\}_{s=1}^{BS}) \\ 0, & \text{otherwise} \end{cases} \tag{6}$$

$$\mathbf{y}_s = \sum_{i=1}^{N} \mathbf{G}_{s,i}^{Dy} E(\mathbf{x}_s). \tag{7}$$

**Capacity and Computational Cost.** To establish a rigorous and fair comparison framework, we define the capacity $C^E$ for a single expert $E$, which serves as a standardized metric for quantifying computational costs. This capacity metric enables fair comparisons between DiffMoE and baseline

models by accurately measuring the computational resources utilized by each expert:

$$C^E = \frac{N \times \# \text{ tokens processed by } E}{\# \text{ all input tokens}} = \frac{NK^E}{BS}, \tag{8}$$

where $K^E$ denotes the number of tokens assigned to expert $E$, $N$ is the number of experts, and $BS$ represents the size of global token pool. Here, we define the capacity $C$ for one forward process for both training and inference phases:

$$C = \frac{1}{LN} \sum_{l=1}^{L} \sum_{i=1}^{N} C^{E_i^l}, \tag{9}$$

where $E_i^l$ denotes $i_{\text{th}}$ expert in the $l_{\text{th}}$ MoE layer. During training phase, $C_{\text{train}}$ is fixed to 1 across all the MoE models, indicating that they keep the same computational cost as dense models. Specifically, DiffMoE keeps $K^E_{\text{train}} = BS/N$ for $C_{\text{train}} = 1$. TC-DiT selects the $\texttt{top-1}$ expert, while EC-DiT selects $\texttt{top-}(S/N)$ tokens per sample in a batch to ensure the same computation. During the inference phase, we compute the global average inference capacity $C_{\text{infer}}^{\text{avg}}$ by averaging over all timesteps: $C_{\text{infer}}^{\text{avg}} = \frac{1}{T} \sum_{t=1}^{T} C_{\text{infer}}^t$, where $C_{\text{infer}}^t$ represents the inference capacity at sampling step $t$.

**Capacity Predictor.** Although batch-level global token routing enables efficient model training, conventional MoE inference strategies with fixed computational resource allocation fail to fully leverage its potential. This limitation stems from the static resource distribution across different noise levels and conditional information during inference. To optimize token selection, we propose a *capacity predictor*—a lightweight structure that dynamically determines token selection per expert through a two-layer MLP with SiLU activations. This adaptive mechanism learns from training-time token routing patterns, efficiently distributing computational resources between complex and simple cases. Formally, let $\text{CP}(\mathbf{x}_{\text{pool}}) \in \mathbb{R}^{BS \times N}$ denote the predictor's output for input $\mathbf{x}_{\text{pool}} \in \mathbb{R}^{BS \times D}$:

$$\text{CP}(\mathbf{x}_{\text{pool}}) = \mathbf{W}_2 \sigma_{\text{SiLU}}(\mathbf{W}_1 \mathbf{x}_{\text{pool}}). \tag{10}$$

We can optimize the capacity predictor by minimizing the object function:

$$\mathcal{L}_{\text{CP}} = BCELoss(\mathbf{O}, \text{CP}(\texttt{sg}[\mathbf{x}_{\text{pool}}])) \tag{11}$$

where $\texttt{sg}$ denotes the $\texttt{stop-gradient}$ operation, and $\mathbf{O} \in \mathbb{R}^{L \times BS \times N}$ is defined as follows:

$$\mathbf{O}_{s,i}^l = \begin{cases} 1, & \text{if } \mathbf{x}_{\text{pool},s} \text{ is processed by } E_i^l \\ 0, & \text{otherwise} \end{cases}. \tag{12}$$

We employ the $\texttt{stop-gradient}$ technique to train the capacity predictor, ensuring it focuses solely on the input features which contains vision, conditions and step information at the current layer while preventing it from interfering with the training of the main diffusion transformers. Therefore, $\mathcal{L}_{\text{CP}}$ will not affect actual diffusion loss. During inference, the capacity predictor determines the inference capacity $C_{\text{infer}}^{E_i^l,t}$ for each expert $E_i^l$ at timestep $t$ based on a threshold. Let $\tau_{E_i^l}$ denote the threshold of $E_i^l$. Using $\mathcal{T} = \{\tau_{E_i^l} \mid i \in \{1, \ldots, N\}, l \in \{1, \ldots, L\}\}$, the model achieves an adaptive $C_{\text{infer}}^{E_i^l,t}$ allocation at sampling step $t$ tailored to different input tokens as follows:

$$C_{\text{infer}}^{E_i^l,t}(\tau_{E_i^l}) = \frac{N}{BS} \sum_{s=1}^{BS} \delta_{s,i}^l \quad \text{where } \delta_{s,i}^l = \begin{cases} 1, & \text{if } \text{CP}(\mathbf{x}_{\text{pool}})_{s,i} > \tau_{E_i^l} \\ 0, & \text{otherwise} \end{cases} \tag{13}$$

Then we can calculate $C_{\text{infer}}^{\text{avg}}$

$$C_{\text{infer}}^{\text{avg}}(\mathcal{T}) = \frac{1}{TNL} \sum_{t=1}^{T} \sum_{i=1}^{N} \sum_{l=1}^{L} C_{\text{infer}}^{E_i^l,t}(\tau_{E_i^l}). \tag{14}$$

**Dynamic Threshold.** We can set the threshold $\mathcal{T}$ to control the number of tokens processed by each expert during inference. It is evident that #tokens processed during inference, decreases as $\tau_{E_i^l}$

increases for all expert. We can adjust $\mathcal{T}$ flexibly to achieve a better trade-off between computational complexity and generation quality. We employ two distinct approaches for threshold $\mathcal{T}$ determination: *Interval Search* and *Dynamic Threshold*. The interval search method addresses an optimization problem formulated as follows:

$$\min_{\mathcal{T}} \text{FID}(\mathcal{T}) \quad \text{subject to } C_{\text{infer}}^{\text{avg}}(\mathcal{T}) \leq 1. \tag{15}$$

To simplify the optimization problem, we assume that $\tau = \gamma < 1, \forall \tau \in \mathcal{T}$, where $\gamma$ is a constant in our experiments. However, interval search method is labor-intensive and time-consuming, making it impractical for real-world applications. To address this limitation, we propose a dynamic threshold method that automatically maintains thresholds (denoted as $\mathcal{T}^{Dy} = \{\tau_{E_i^l}^{Dy} \mid i \in \{1, \ldots, N\}, l \in \{1, \ldots, L\}\}$) for all experts during the training phase. To ensure the inference computational cost approximates the training cost (i.e. $C_{\text{infer}}^{\text{avg}} \approx 1$), we employ the Exponential Moving Average (EMA) technique as follows:

$$\text{Quantile}_{E_i^l} \leftarrow \text{CP}(\mathbf{x}_{\text{pool}})_{s_k, i},$$
$$\tau_{E_i^l}^{Dy} \leftarrow \alpha \cdot \tau_{E_i^l}^{Dy} + (1 - \alpha) \cdot \text{Quantile}_{E_i^l}, \tag{16}$$

where $s_k$ denotes the $k_{\text{th}}$ value in descending order, $\alpha$ is a constant which is equals to 0.95 in our experiments.

## 3 EXPERIMENTS

We evaluate DiffMoE on class-conditional and text-to image generation along three key dimensions:

(1) **Training and Inference Performance** Section 3.2. Tables 1, 2, 3, 4, 5, 12, and 14. Figure 5.

(2) **Dynamic Computation** Section 3.3. Figure 1, 3, 7, 8. Table 15. Appendix B.

(3) Section 3.4. Table 2, 3, 4, 5. Figure 4, 6.

For the convenience of elaboration, we use the flow matching training method to do the following analysis while DDPM results are also provided in the Appendix A.1.

### 3.1 EXPERIMENT SETUP

**Baseline and Model architecture.** We compare DiffMoE against Dense-DiT, TC-DiT (Fei et al., 2024), and EC-DiT (Sun et al., 2024) using both flow matching (Ma et al., 2024) and denoising score matching (Peebles & Xie, 2023a). All models follow the naming convention **[Model]-[Size]-[#Experts]-[Type]**. For class-conditional generation, we replace even FFN layers with MoE layers while maintaining the original DiT architecture (details in Appendix Table 9). For text-to-image generation, we incorporate cross-attention modules (Rombach et al., 2022) to introduce text conditions. In this setup, all the models activates 458M parameters and the total parameters of MoE variants are 1.2B. Further training details are provided in Appendix D.1.

**Evaluation.** We evaluated DiffMoE through both quantitative and qualitative metrics. Quantitatively, we used FID50K (Heusel et al., 2017) with 250 DDPM/Euler (250 NFEs) steps for class-conditional generation, For text-to-image generation evaluation, we use FID (Heusel et al., 2017), CLIP Score (Radford et al., 2021), PickScore (Kirstain et al., 2023), and HPSv2 (Wu et al., 2023) on COCO prompts. For PickScore and HPSv2, we compute the average score and standard deviation across prompts, ensuring fair comparison using identical prompt-image pairs. We report performance on the GenEval (Ghosh et al., 2023) and T2I-CompBench (Huang et al., 2023) to evaluate all-round capabilities of DiffMoE and baselines following their official protocol.

### 3.2 MAIN RESULTS OF C2I AND T2I

**C2I Results.** Class-conditional image generation is a task of synthesizing images based on specified class labels. DiffMoE-L-E16 demonstrates superior efficiency by outperforming Dense-DiT-XL (which uses $1.5\times$ more parameters) after just 700K steps, as shown in Table 1. And these improvements hold across both the DDPM and Flow Matching paradigms. With extended training for 3000K

Table 1: **Baseline and Capacity Predictor Effects.** Performance comparison of different model architectures at 700K steps. DiffMoE-L-E16-Flow achieves best FID50K (14.41 w/o CFG) among TC, EC, and Dense variants. $\times$ means n times base activated parameters. A.A.P indicates Average Activated Params. **Bold** indicates the best result, and underline indicates the second best result.

| Model (700K) | Config | # A.A.P | FID50K$\downarrow$ |
|---|---|---|---|
| TC-DiT-L-E16-Flow | Token-Choice MoE | 458M $\times$ | 19.06 |
| EC-DiT-L-E16-Flow | Expert-Choice MoE | 458M $\times$ | 16.12 |
| Dense-DiT-L-Flow | Dense FFN | 458M $\times$ | 17.01 |
| Dense-DiT-XL-Flow | Dense FFN | 675M 1.5$\times$ | 14.77 |
| **DiffMoE-L-E16-Flow** | Token Pool Only | 458M $\times$ | 15.25 |
| **DiffMoE-L-E16-Flow** | Token Pool & Predictor | 454M 0.95$\times$ | **14.41** |

Table 2: **Parameter Scaling Behavior** on ImageNet 256$\times$256 Class-Conditional Generation. DiffMoE demonstrates superior FID scores with fewer parameters after 3000K training steps. Results are reported with guidance scale 1.5.

| Diffusion Models (3000K) | # A.A.P | FID50K$\downarrow$ | IS$\uparrow$ | Precision$\uparrow$ | Recall$\uparrow$ |
|---|---|---|---|---|---|
| Dense-DiT-XL-Flow | 675M 1.5$\times$ | 2.52 | 273.78 | 0.84 | 0.56 |
| Dense-DiT-XXL-Flow | 951M 2$\times$ | 2.41 | 281.96 | 0.84 | 0.57 |
| Dense-DiT-XXXL-Flow | 1353M 3$\times$ | 2.37 | 291.29 | 0.84 | 0.57 |
| **DiffMoE-L-E8-Flow** | 458M $\times$ | 2.40 | 280.30 | 0.83 | 0.57 |
| **DiffMoE-L-E16-Flow** | 458M $\times$ | 2.36 | 287.26 | 0.83 | 0.58 |
| **DiffMoE-XL-E16-Flow** | 675M 1.5$\times$ | **2.30** | 291.23 | 0.83 | 0.58 |

steps, DiffMoE-L-E16 (with 1$\times$ parameters and FID 2.36) surpasses Dense-DiT-XXXL (with 3$\times$ parameters and FID 2.37), as shown in Table 2. Visualizations are provided in Figures 10 and 11.

**T2I Results.** In text-to-image generation, DiffMoE consistently outperforms dense and moe variants baselines in various metrics which can evaluate multi-dimensions include visual fidelity, diversity, text-image alignment and prompt understanding. Results are shown in 3, 4, 5.

## 3.3 DYNAMIC COMPUTATION ANALYSIS

**Analysis of Inference Capacity.** DiffMoE-L-E16-Flow demonstrates superior parameter efficiency with its inference capacity ($C_{\text{infer}}^{\text{avg}}$) being 1 less than TC-DiT and EC-DiT, while achieving better performance, as shown in Table 1. Notably, with only 454M(0.95$\times$) average activated parameters, our model outperforms Dense-DiT-XL-Flow (675M 1$\times$), highlighting the effectiveness of dynamic token allocation. Detailed analysis of average activated parameters is provided in the Appendix E.

**Ablation of Capacity Predictor.** Dynamic token selection through our capacity predictor demonstrates superior performance over traditional static topK token selection, as shown in Figure 3(a) and Table 1. This improvement stems from the predictor's ability to intelligently allocate more computational resources to challenging tasks. The capacity predictor plays a crucial role in unleashing DiffMoE's full potential by dynamically adjusting resource allocation, which is particularly important for optimizing inference efficiency.

**Interval Search vs. Dynamic Threshold.** Both interval search and dynamic threshold methods achieve optimal performance in DiffMoE, with the dynamic threshold ($\mathcal{T}^{Dy}$) emerging as our preferred approach due to its elegance and efficiency. Through interval search from 0.0 to 0.999, we identify an optimal threshold ($\gamma \approx 0.4$) that minimizes FID while maintaining $C_{\text{infer}}^{\text{avg}} \leq 1$. Meanwhile, the dynamic threshold automatically maintains $C_{\text{infer}}^{\text{avg}} \approx 1$ during inference, achieving comparable FID scores within the optimal region, as shown in Figure 3(b) and Table 11. Our experiments reveal a U-shaped relationship between FID and $C_{\text{infer}}^{\text{avg}}$, indicating that both over-activation and under-activation of parameters degrade performance. Both methods successfully identify thresholds within

Table 3: **Text-to-Image results on COCO prompts.** Under the same 458M activated parameters, DiffMoE consistently outperforms all baseline DiT variants across various T2I metrics.

| Model | FID10K↓ | CLIP Score↑ | PickScore↑ | HPSv2↑ |
|-------|---------|-------------|------------|--------|
| Dense-DiT | 46.38 | 29.58 | 0.2510±0.111 | 0.255±0.021 |
| TC-DiT | 45.10 | 29.59 | 0.2024±0.112 | 0.256±0.020 |
| EC-DiT | 44.80 | 29.60 | 0.2527±0.109 | 0.255±0.020 |
| **DiffMoE** | **44.53** | **29.75** | **0.2937±0.112** | **0.257±0.020** |

Table 4: **Text-to-Image GenEval Benchmark Results.** DiffMoE outperforms baseline DiT variants across six evaluation aspects, showing better compositionality, faithfulness, and controllability.

| Model | Single Obj.↑ | Two Obj.↑ | Cnt. Obj.↑ | Colors↑ | Pos.↑ | Attri.↑ | Overall↑ |
|-------|--------------|-----------|------------|---------|-------|---------|----------|
| Dense-DiT | 0.656 | 0.280 | **0.225** | 0.481 | 0.165 | 0.278 | 0.346 |
| TC-DiT | 0.644 | 0.298 | 0.194 | **0.500** | 0.178 | 0.275 | 0.348 |
| EC-DiT | 0.666 | 0.349 | 0.181 | 0.426 | **0.200** | 0.235 | 0.343 |
| **DiffMoE** | **0.688** | **0.397** | 0.200 | **0.500** | 0.178 | **0.303** | **0.377** |

the optimal region, but the dynamic threshold's straightforward implementation and computational efficiency make it our default choice throughout this paper.

**Harder Work Needs More Computation.** Figure 1(c) illustrates that different classes demand varying amounts of computation during generation. By analyzing 1K class labels and ranking their $C_{\text{infer}}^{\text{avg}}$, we observe clear patterns in computational requirements. The most challenging cases often involve objects with intricate details, complex materials, precise structures, or specific viewing angles (e.g., technical instruments and detailed artifacts). In contrast, natural subjects such as common animals (e.g., birds, dogs, cats) generally require less computation. Figures 7 and 8 present the top-10 most and least computationally demanding classes for both flow-based and DDPM models, respectively. For text-to-image generation, we also identify harder and easier prompts by similiar ranking method. Detailed discussions refer to Appendix B.1.

### 3.4 SCALING BEHAVIOR

**Scaling Model Size.** DiffMoE demonstrates consistent performance improvements across small (S), base (B), and large (L) configurations, with activated parameters of 32M, 130M, and 458M respectively as shown in Figure 4(a). To explore the upper limits of DiffMoE and quantify its performance efficiency, as analyzed in Section 3.2 DiffMoE-L-E16 (with $1\times$ parameters and FID 2.36) surpasses Dense-DiT-XXXL (with $3\times$ parameters and FID 2.37) with same 3000K training budget. This highlights the exceptional parameter efficiency and scalability of DiffMoE.

**Scaling Number of Experts.** As shown in Figure 4(b), model performance improves consistently when scaling experts from 2 to 16, with diminishing returns between E8 and E16. Based on this analysis, we trained DiffMoE-L-E8 for 7000K iterations, achieving optimal performance-efficiency trade-off and state-of-the-art results.

### 3.5 LIMITATIONS AND BROADER IMPACT

While DiffMoE demonstrates promising results, our study has two limitations. First, due to computational constraints, we have not validated the framework on text-to-video generation tasks, leaving this extension for future work. Second, like other generative models, DiffMoE could potentially be misused to create harmful content, warranting careful consideration of ethical deployment.

## 4 RELATED WORKS

**Diffusion Models.** Diffusion models (Ho et al., 2020; Podell et al., 2023; Peebles & Xie, 2023a; Esser et al., 2024b) have emerged as the dominant paradigm in visual generation in recent years.

Table 5: **Text-to-Image T2I-CompBench Results.** DiffMoE consistently outperforms baseline DiT variants across color, texture, shape, number, spatial, and complex compositional aspects.

| Model | Color ↑ | Texture ↑ | Shape ↑ | Number ↑ | 3D-Spatial ↑ | 2D-Spatial ↑ | Non-Spatial ↑ | Complex ↑ |
|---|---|---|---|---|---|---|---|---|
| Dense-DiT | 0.598 | 0.485 | 0.320 | 0.373 | 0.261 | 0.0163 | 0.292 | 0.292 |
| TC-DiT | 0.582 | 0.519 | 0.338 | 0.377 | 0.256 | 0.0178 | 0.290 | 0.293 |
| EC-DiT | 0.598 | 0.532 | 0.369 | 0.335 | 0.242 | 0.0205 | 0.293 | 0.302 |
| **DiffMoE** | **0.601** | **0.547** | **0.395** | **0.432** | **0.264** | **0.0234** | **0.295** | **0.311** |

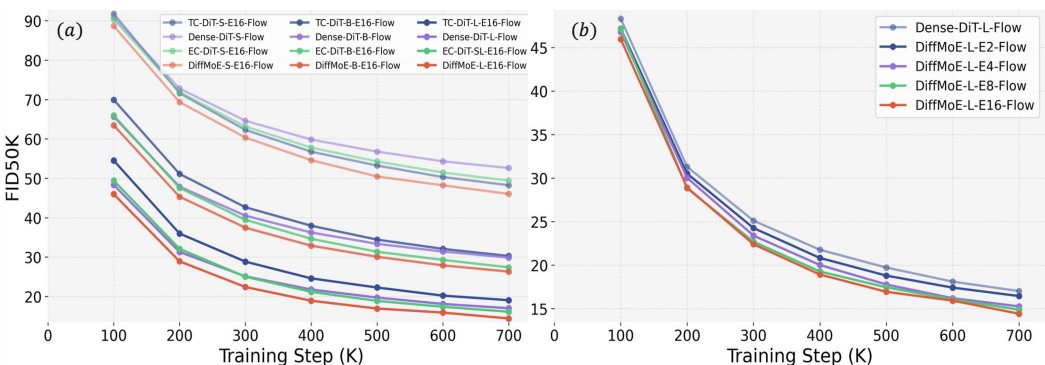

Figure 4: **Scaling Analysis. (a) Scaling Model Size:** DiffMoE consistently outperforms the corresponding baseline models across all scales (Small/Medium/Large). **(b) Scaling Number of Experts :** Comparison of FID50K scores during training between Dense-DiT-L-Flow (E1) and models with increasing expert counts (E2, E4, E8, E16).

These models transform gaussian distribution into target data distribution through iterative processes, with two primary training paradigms: Denoising Diffusion Probabilistic Models (DDPM) trained via score-matching (Ho et al., 2020; Song et al., 2021), which learns the inverse of a diffusion process and Rectified Flow approaches optimized through flow-matching (Lipman et al., 2022; Ma et al., 2024; Esser et al., 2024b), which is a more generic modeling techinique and can construct a straight probaility path connecting data and noise. We implement DiffMoE using both paradigms, demonstrating its versatility across these complementary training methodologies.

**Mixture of Experts.** Mixture of Experts (MoE) (Shazeer et al., 2017; Lepikhin et al., 2020) enables efficient model scaling through conditional computation by selectively activating expert subsets, and has shown remarkable success in LLMs such as DeepSeek-V3 (DeepSeek-AI et al., 2024). Recent attempts to incorporate MoE into diffusion models face several limitations. MEME (Lee et al., 2023), eDiff-I (Balaji et al., 2022), and ERNIE-ViLG 2.0 (Feng et al., 2023) restrict experts to specific timesteps; SegMoE (Yatharth Gupta, 2024) and DiT-MoE (Fei et al., 2024) suffer from expert imbalance due to isolated token processing; and EC-DiT (Sehwag et al., 2024; Sun et al., 2024) limits token selection to individual samples and requires longer training for marginal gains. By constraining global token distribution across noise levels and conditions, these methods fail to capture the inherent heterogeneity of diffusion processes. Other related works include Diff-MoE (Cheng et al.), which employs time- and space-adaptive expert routing, and DyDiT (Zhao et al., 2024), which focuses on dynamic inference computation; both are orthogonal to our batch-level token routing and scalable architecture. DiffMoE addresses the above challenges through a batch-level global token pool for training and computation that adapts to noise levels and sample complexity during inference.

## 5  CONCLUSION

In this work, we propose DiffMoE, a simple yet effective framework for scaling diffusion models via dynamic token selection and global token accessibility. By overcoming the uniform processing bottleneck of diffusion transformers, DiffMoE achieves superior performance over TC-MoE, EC-MoE, and even dense models with 3× parameters, while keeping computational costs comparable. Our results highlight its broad applicability to both class-conditional and text-to-image generation, positioning DiffMoE as a scalable and efficient foundation for advancing large-scale diffusion models.

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

# A MORE DIFFMOE ANALYSIS

## A.1 ADDITIONAL DIFFMOE DDPM CLASS-CONDITIONAL GENERATION QUALITATIVE AND QUANTITATIVE RESULTS

We present comprehensive evaluations of DiffMoE-L-E16-DDPM series models. Table 12 shows the experimental results, while Figure 5 illustrates the diffusion loss comparison against the baseline model, revealing substantial performance improvements. Furthermore, Figure 6 demonstrates the scaling capabilities of our DiffMoE-DDPM architecture.

## A.2 COMPUTATIONAL OVERHEAD AND HARDWARE PERFORMANCE ANALYSIS

### A.2.1 COMPUTATIONAL OVERHEAD ANALYSIS

DiffMoE maintains high training efficiency while introducing minimal overhead. Compared with the cost of a standard feed-forward network (FFN), the computational cost of the MoE router is negligible. The primary overhead arises from the top-$K$ selection operation, whose complexity grows logarithmically with respect to the number of tokens or experts. Additionally, token indexing operations (e.g., `y[mask] = gate[mask] * x[mask]` to gather routed tokens) can introduce extra computational cost compared with dense models if implemented naively.

Table 6: Computation Complexity Analysis

| MoE Method | Top-K Complexity | FFN Complexity | Relative Overhead | Token Indexing | Training Speed (steps/sec) |
|---|---|---|---|---|---|
| Dense-DiT | N/A | $\mathcal{O}(BSD^2)$ | N/A | N/A | 1.9 |
| TC-MoE | $\mathcal{O}(BSE\log(E))$ | $\mathcal{O}(BSD^2)$ | $10^{-5}$ (negligible) | Required | 1.5 |
| EC-MoE | $\mathcal{O}(BSE\log(S))$ | $\mathcal{O}(BSD^2)$ | $10^{-3}$ | Required | 1.6 |
| **DiffMoE** | $\mathcal{O}(BSE\log(BS))$ | $\mathcal{O}(BSD^2)$ | $10^{-2}$ (minor) | Required | 1.6 |

*Note:* $B$ denotes batch size, $S$ sequence length, $D$ embedding dimension, and $E$ number of experts.

### A.2.2 HARDWARE-RELATED PERFORMANCE METRICS

We benchmarked training and inference on NVIDIA H800 GPUs using Fully Sharded Data Parallel (FSDP). Training was conducted with a batch size of 64, while inference used a single GPU with batch size 20. Input resolution was fixed at $256 \times 256$ pixels.

Table 7: Training Time Hardware-related Performance Metrics

| Model | Active Params (M) | Training Speed (iters/sec) | Training Memory Consumption (GB) |
|---|---|---|---|
| Dense-DiT | 458 | 1.9 | 29 |
| TC-DiT | 458 | 1.5 | 31 |
| EC-DiT | 458 | 1.6 | 31 |
| **DiffMoE** | 458 | 1.6 | 31 |

Table 8: Inference Time Hardware-related Performance Metrics

| Model | Active Params (M) | Throughput (images/sec) | Latency (sec/image) | Inference Speed (step/sec) |
|---|---|---|---|---|
| Dense-DiT | 458 | 6.57 | 0.152 | 8.22 |
| TC-DiT | 458 | 4.8 | 0.208 | 6 |
| EC-DiT | 458 | 5.75 | 0.173 | 7.19 |
| **DiffMoE** | 458 | 5.74 | 0.174 | 7.18 |

All sparse models maintain the same number of active parameters as the dense baseline, ensuring a fair comparison. While a slight increase in memory usage (approximately 2GB) and a modest decrease in throughput are observed, these overheads are minor and do not affect DiffMoE's deployment feasibility. MoE models can be further improved by advanced frameworks like Megatron Shoeybi et al. (2019).

Table 9: **DiffMoE Model Configurations**. Hyperparameter settings and computational specifications for class-conditional models. See Appendix E for activated parameter calculations.

| Model Config | #Avg. Activated Params ($C_{\text{infer}}^{\text{avg}} = 1$). | #Total Params. | #Blocks L | #Hidden dim. D | #Head n | #Experts | $C_{\text{train}}$ |
|---|---|---|---|---|---|---|---|
| DiffMoE-S-E16 | 32M | 139M | 12 | 384 | 6 | 16 | 1 |
| DiffMoE-B-E16 | 130M | 555M | 12 | 768 | 12 | 16 | 1 |
| DiffMoE-L-E8 | 458M | 1.176B | 24 | 1024 | 16 | 8 | 1 |
| DiffMoE-L-E16 | 458M | 1.982B | 24 | 1024 | 16 | 16 | 1 |
| DiffMoE-XL-E16 | 675M | 2.925B | 28 | 1152 | 16 | 16 | 1 |

Table 10: **Module Parameters and Percentage.** We have counted the number of parameters (M) of various modules to facilitate our analysis.

| Model | FFN | Attention | AdaLN | Others | Total |
|---|---|---|---|---|---|
| Dense-DiT-L | 201.44(44.0%) | 100.7(22.0%) | 151(33.0%) | 4.7(1.0%) | 457.84 |
| **DiffMoE-L-E2** | 314.8(55.1%) | 100.7(17.6%) | 151(26.4%) | 4.7(0.8%) | 571.2 |
| **DiffMoE-L-E4** | 516.3(66.8%) | 100.7(13.0%) | 151(19.5%) | 4.7((0.6%) | 772.7 |
| **DiffMoE-L-E8** | 919.3(78.2%) | 100.7(8.6%) | 151(12.8%) | 4.7(0.4%) | 1175.7 |
| **DiffMoE-L-E16** | 1725.3(87.1%) | 100.7(5.1%) | 151(7.6%) | 4.7(0.2%) | 1981.7 |

Table 11: **Different Threshold Method.** We use both interval search and dynamic threshold method to find out the optimal $\tau_{E_i}$. We find that the dynamic threshold makes a good balance between $C_{\text{infer}}^{\text{avg}}$ and performance.

| $\mathcal{T}$ | $C_{\text{infer}}^{\text{avg}}$ | FID50K ↓ |   | $\mathcal{T}$ | $C_{\text{infer}}^{\text{avg}}$ | FID50K ↓ |
|---|---|---|---|---|---|---|
| 0.999 | 0.41 | 36.28 |   | 0.2 | 1.10 | 13.38 |
| 0.99 | 0.51 | 24.22 |   | 0.1 | 1.22 | 12.92 |
| 0.9 | 0.71 | 17.59 |   | 1E-2 | 1.71 | 12.14 |
| 0.8 | 0.78 | 16.21 |   | 1E-3 | 2.33 | **11.82** |
| 0.7 | 0.83 | 15.51 |   | 1E-4 | 3.17 | 11.87 |
| 0.6 | 0.88 | 15.00 |   | 1E-5 | 4.36 | 12.47 |
| 0.5 | 0.92 | 14.59 |   | 1E-6 | 6.01 | 13.51 |
| 0.4 | 0.97 | **14.16** |   | 1E-7 | 7.88 | 13.96 |
| 0.3 | 1.03 | 13.75 |   | 1E-8 | 9.67 | 16.85 |
| 0.2 | 1.10 | 13.38 |   | 1E-9 | 11.18 | 18.90 |
| 0.1 | 1.22 | 12.92 |   | 0.0 | 16 | 29.39 |
| Dynamic | 0.95 | 14.41 |   |   |   |   |

Table 12: **Comparisons with the Baseline Models. (DDPM)** We compare TC, EC and Dense Model and show the average activated parameters of all the experts across all the sampling steps.

| Model (700K) | # Avg. Activated Params. | $C_{\text{infer}}^{\text{avg}}$ | FID50K ↓ |
|---|---|---|---|
| TC-DiT-L-E16-DDPM | 458M | 1 | 20.81 |
| EC-DiT-L-E16-DDPM | 458M | 1 | 17.65 |
| Dense-DiT-L-DDPM | 458M | 1 | 17.87 |
| Dense-DiT-XL-DDPM | 675M | 1 | 15.28 |
| **DiffMoE-L-E16-DDPM**4 | 458M | 1 | **14.60** |

Table 13: **Decoder Ablation Study**. Evaluation of various pre-trained VAE decoder weights. †: results from Ma et al. (2024) (DDPM) and Peebles & Xie (2023a) (Flow). *: our reproduction. All other results are from our experiments. In general, with VAE decoder EMA version, the FID score is consistently lower than MSE version.

| Model | Training Steps | VAE-Decoder | Sampler | Batch Size | FID50K ↓ |
|---|---|---|---|---|---|
| Dense-DiT-XL-Flow | 400K | ft-MSE | Euler | 125 | 18.80 |
| Dense-DiT-XL-Flow | 400K | ft-EMA | Euler | 125 | 18.74 |
| Dense-DiT-XL-Flow | 400K | ft-MSE | Dopri5 | 125 | 18.63 |
| Dense-DiT-XL-Flow | 400K | ft-EMA | Dopri5 | 125 | **18.45** |
| Dense-DiT-XL-Flow* | 7000K | ft-MSE | Heun | 125 | 9.66 |
| Dense-DiT-XL-Flow* | 7000K | ft-EMA | Heun | 125 | 9.63 |
| Dense-DiT-XL-Flow* | 7000K | ft-MSE | Dopri5 | 125 | 9.51 |
| Dense-DiT-XL-Flow* | 7000K | ft-EMA | Dopri5 | 125 | **9.48** |
| Dense-DiT-XL-DDPM-G † | 7000K | ft-MSE | DDPM | 125 | 2.30 |
| Dense-DiT-XL-DDPM-G † | 7000K | ft-EMA | DDPM | 125 | **2.27** |

## A.3 ANALYSIS OF TOKEN INTERACTION STRATEGIES.

As shown in Figure 1. The interaction levels (L1/L2/L3) represent: L1 for isolated token processing, L2 for local token routing within samples, and L3 for global token routing across sample. Table 14 presents a comprehensive comparison of different token interaction strategies in diffusion models. The baseline models with L1 strategy (TC-DiT-L-Flow and Dense-DiT-L-Flow) process tokens independently, resulting in limited performance (FID: 19.06 and 17.01). The L2 strategy, implemented in EC-DiT-L-Flow, enables local token routing within samples, showing improved performance (FID: 16.12) with the same parameter count. Our proposed L3 strategy in DiffMoE-L-Flow introduces cross-sample token routing, achieving superior results (FID: 14.41) even compared to the 1.5x larger Dense-DiT-XL-Flow (675M parameters). Notably, when combined with Dynamic Global CP, our model not only achieves the best FID score but also reduces the computational capacity to 0.95x, demonstrating both effectiveness and efficiency.

## B DYNAMIC CONDITIONAL COMPUTATION

### B.1 HARDER WORK NEEDS MORE COMPUTATION

**Dynamic Computation in C2I:** Figure 1 demonstrates that different classes require varying computational resources during generation. To analyze this variation, we sample 1K different class labels in a batch and rank their $C_{\text{infer}}^{\text{avg}}$ in descending order, revealing the computational complexity of generation across classes. The top-10 most computationally intensive classes for both flow-based and DDPM models are displayed in Figure 7. The top-10 least computationally intensive classes for both flow-based and DDPM models are displayed in Figure 8.

(a) Loss Comparison of L-Flow Series      (b) Loss Comparison of L-DDPM Series

Figure 5: **Loss Comparison of L-Flow and L-DDPM Series**. The relative losses illustrated in subfigures (a) and (b) demonstrate the exceptional training dynamics of DiffMoE, consistently outperforming all baseline models.

Table 14: **Performance comparison of different diffusion models with varying token interaction strategies**. All models are trained with Flow Matching for 700K steps. The interaction levels (L1/L2/L3) represent: L1 (Iso) for isolated token processing, L2 (Loc) for local token routing within samples, and L3 (Glob) for global token routing across samples. Our DiffMoE-L-Flow with Dynamic Global CP achieves the best FID score of 14.41 while maintaining parameter efficiency and reduced computational cost. # A.A.P. denotes # Avg. Act. Params.

| Model | # A.A.P. | Train | Infer | FID50K $\downarrow$ |
|---|---|---|---|---|
| TC-DiT-L-E16-Flow | 458M | L1 (Isolated) | L1 (Isolated) | 19.06 |
| Dense-DiT-L-Flow | 458M | L1 (Isolated) | L1 (Isolated) | 17.01 |
| EC-DiT-L-E16-Flow | 458M | L2 (Local) | L2 (Loc, Stat TopK) | 16.12 |
| EC-DiT-L-E16-Flow | 458M | L2 (Local) | L2 (Loc, Dyn Intra Sample) | 23.74 |
| DiffMoE-L-E16-Flow | 458M | L3 (Global) | L3 (Glob, Stat TopK) | 15.25 |
| Dense-DiT-XL-Flow | 675M | L1 (Isolated) | L1 (Isolated) | 14.77 |
| DiffMoE-L-E16-Flow | 454M | L3 (Global) | L3 (Glob, Dyn Cross Sample) | **14.41** |

**Dynamic Computation in T2I:**

The dynamic computation paradigm has also been explored in text-to-image (T2I) generation tasks. To investigate its effectiveness, we used 10K text prompts sampled from GPT-4o to synthesize images for evaluation. We measured and ranked the per-sample inference capacity $\mathcal{C}$, which reflects the computational demand. By analyzing the distribution of $\mathcal{C}$, we identify distinctive features of both challenging and straightforward samples in text-to-image tasks.

**Hard Samples** typically exhibit the following characteristics:

- **Fine-grained text rendering:** e.g., a signboard with multiple handwritten fonts, or a calendar with circled dates and notes.
- **Complex spatial relations:** e.g., objects arranged in layered or interleaved configurations, such as furniture in a cluttered room.
- **Multi-object interactions:** e.g., a group of people passing a basketball, or multiple bowls each containing different items.
- **Material fidelity and surface effects:** e.g., shiny or translucent surfaces like glass cups, glossy donuts, or metallic reflections.

**Easy Samples** are characterized by:

- **Simple compositions:** e.g., a single flower, or one object on a plain background.
- **Clear and minimal semantics:** e.g., a sunset over the ocean, or a green apple on a white table.

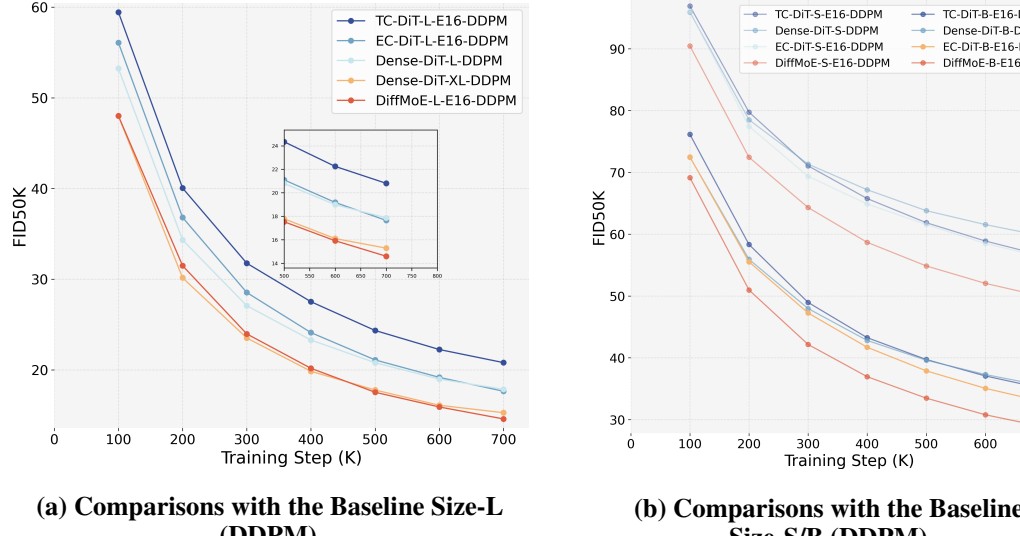

**(a) Comparisons with the Baseline Size-L (DDPM).**

**(b) Comparisons with the Baseline Size-S/B (DDPM).**

Figure 6: **Comparisons with the Baseline Models. (a)** We compare TC, EC, and Dense Models and show the average activated parameters of all experts across all sampling steps. DiffMoE-L-E16-DDPM even surpasses DenseDiT-XL-DDPM (1.5x params). **(b)** We also examine S/B size DiffMoE models to further demonstrate the scalability.

- **Simple attribute emphasis:** e.g., a red book, or a yellow balloon.

Our method effectively adapts computation allocation based on such sample-level difficulty, thereby improving generation quality without requiring manually defined classes or handcrafted heuristics. These findings demonstrate the benefit of dynamic token routing in adapting to varying sample difficulty, providing insights for future research on adaptive computation in generative models. Table 15 provides concrete examples illustrating the correlation between prompt characteristics and computational difficulty.

Table 15: Representative text prompts across different difficulty levels. $C_{\text{infer}}^{\text{avg}}$ denotes the average inference capacity, where higher values indicate greater computational demand.

| Key Feature | Example Prompt | Difficulty | $C_{\text{infer}}^{\text{avg}}$ |
|---|---|---|---|
| Text rendering; complex spatial relations | *A close-up view of a calendar. A specific date, the 20th, is circled and has the word "Branch!" written next to it. A pen is positioned on top of the calendar, pointing towards the circled date.* | Hard | 1.13 |
| Multi-object; complex spatial relations | *Three wooden bowls placed side by side on a textured surface. Each bowl contains a different type of legume.* | Hard | 1.07 |
| Simple object; single color | *A vibrant pink flower with delicate petals.* | Easy | 0.92 |
| Simple composition; attribute emphasis | *A heart-shaped padlock with a shiny silver keyhole, set against a vibrant pink background.* | Easy | 0.93 |

## B.2 EXPERT SPECIALIZATION ANALYSIS

Our core finding is that expert specialization in DiffMoE is not random but follows a **systematic depth-dependent progression**. This specialization is directly enabled by the global token pool, which provides each expert with a diverse and representative sample of the current batch's computational demands, allowing them to learn distinct temporal roles. The visualization of expert capacity utilization across timesteps and layers (Figure 9) reveals this evolutionary trajectory:

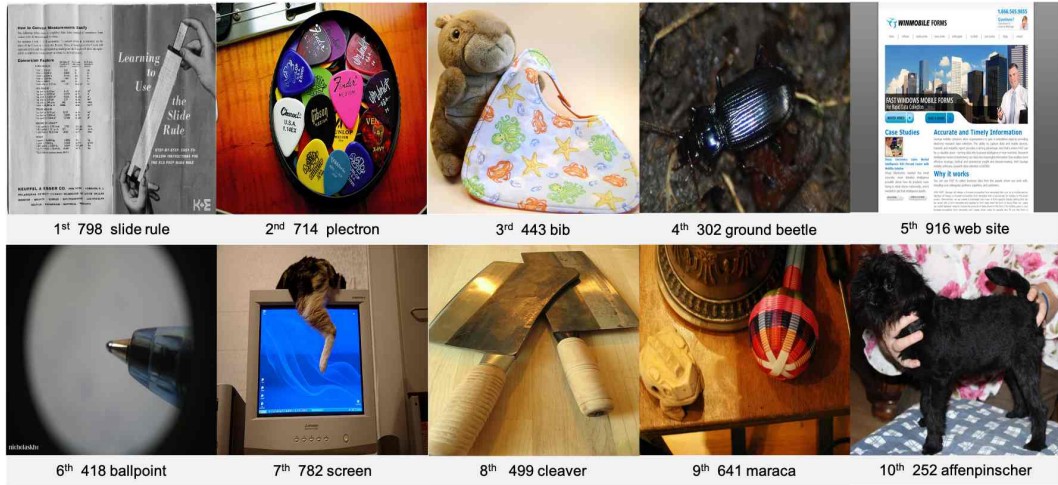

**(a) DiffMoE-L-E16-Flow (700K)**

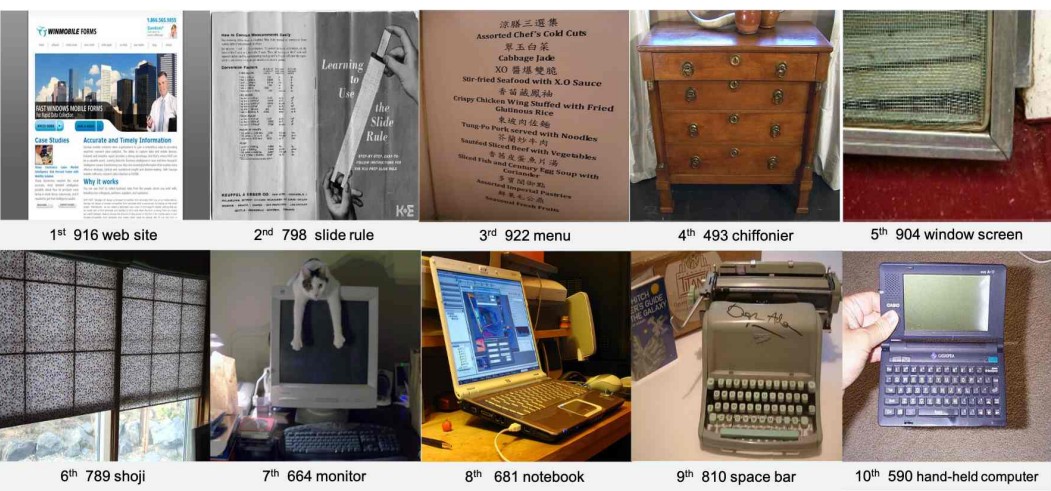

**(b) DiffMoE-L-E16-DDPM (700K)**

Figure 7: **Top 10 Hardest Classes.** The 10 classes with the highest computational cost, sampled from the training set.

- **Layer 1 (shallow)**: Experts act as dynamic "generalists," exhibiting the **"Bookends"** pattern with high capacity utilization during both high-noise (early) and low-noise (late) steps. This indicates their dual role in initial noisy feature parsing and final detail refinement.
- **Layer 7 (intermediate)**: Routing becomes more stable and balanced. Experts here predominantly follow the **"Mid-Peak"** pattern, concentrating computation on the semantically complex mid-range timesteps.
- **Layer 13 (deep-mid)**: Experts show smooth, long-range capacity transitions, handling fewer early-step tokens and focusing more on structured representations emerging in middle and later timesteps.
- **Layer 19 (deep)**: Deep-layer experts demonstrate highly stable and specialized behavior, primarily as **"Anchors"** with consistent engagement. They focus on processing structurally coherent representations during the critical middle-to-late stages, specializing in high-level semantic refinement.

From macro-view of all experts, we also observe a consistent global trend: **experts tend to receive more tokens in middle timesteps**, aligning with our Figure 1(b) and findings reported in SD3

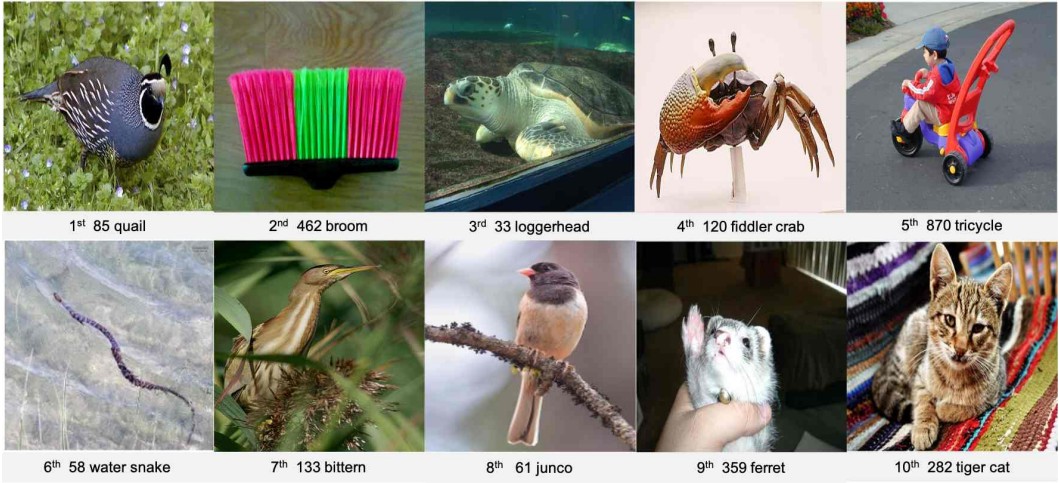

(a) DiffMoE-L-E16-Flow (700K)

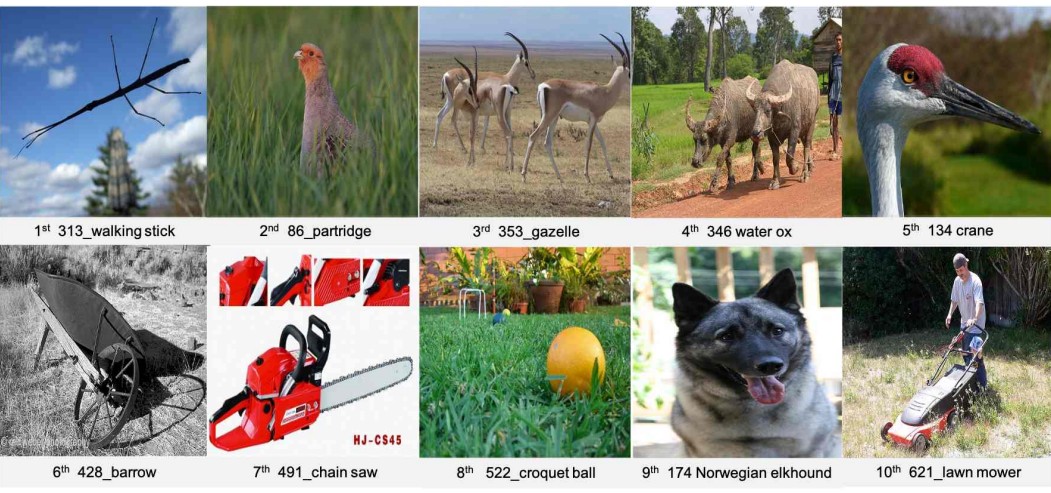

(b) DiffMoE-L-E16-DDPM (700K)

Figure 8: **Top 10 Easiest Classes.** The 10 classes with the lowest computational cost, sampled from the training set.

Esser et al. (2024c) regarding the difficulty of mid-range timesteps for velocity prediction. These observations confirm that **experts do not operate uniformly but gradually specialize**, enabled by the batch-level global token pool and capacity predictor. By exposing experts to a diverse and globally aggregated set of tokens at each step, the global pool encourages specialization toward different noise levels and semantic complexities.

### B.2.1 A Taxonomy of Expert Temporal Specialization

We further categorize the observed expert behaviors into five distinct temporal specialization patterns, which are systematically distributed across the network depth:

1. **The Bookends:** Highly activated at the start and end of the denoising process.
2. **The Mid-Peak:** Most active during the middle stages.
3. **The Crescendo:** Activity gradually increases over time.
4. **The Diminuendo:** Activity gradually decreases over time.
5. **The Anchor:** Maintains a consistent level of activity throughout all steps.

### B.2.2 Summary of Expert Behavior by Layer

Table 16: Summary of Expert Behavior by Layer(Shallow,Mid,Deep-mid, Deep

| Layer | Predominant Routing Pattern(s) | Inferred Specialization |
|---|---|---|
| Layer 1 | The Bookends (Primary), The Mid-Peak | Low-level, noise-robust feature extraction |
| Layer 7 | The Mid-Peak (Primary), The Crescendo, The Diminuendo | Mid-level feature integration and structuring |
| Layer 13 | The Mid-Peak (Primary), The Crescendo, The Anchor | Refined structural and conceptual representation |
| Layer 19 | The Mid-Peak, The Crescendo, The Anchor (Primary) | High-level semantic refinement and detail polishing |

**Takeaway:**The global token pool is the cornerstone of this architecture. By providing a global view of the batch's computational needs at each denoising step, it naturally encourages experts to **diversify and specialize** across the noise spectrum and network hierarchy, forming an efficient, adaptive computational continuum.

## C DETAILED BACKGROUND OF GENERATIVE MODELING.

In this section, we will provide a detailed bachground of generative modeling of both DDPM Ho et al. (2020); Song et al. (2021); Rombach et al. (2022) and Rectified Flow (Lipman et al., 2023; Ma et al., 2024; Esser et al., 2024b) which is helpful to understand the difference and relationship between them.

Generative modeling essentially defines a mapping between $\mathbf{x}_1$ from a noise distribution $p_1(\mathbf{x})$ to $\mathbf{x}_0$ from a data distribution $p_0(\mathbf{x})$ leads to time-dependent processes represented as below

$$\mathbf{x}_t = \alpha_t \mathbf{x}_0 + \sigma_t \epsilon, t \in [0, 1], \tag{17}$$

where $\alpha_t$ is a decreasing function of $t$ and $\sigma_t$ is an increasing function of $t$. We set $\alpha_0 = 1, \sigma_0 = 0$ and $\alpha_1 = 0, \sigma_0 = 1$ to make the marginals $p_t(\mathbf{x}_t) = \mathbb{E}_{\epsilon \sim \mathcal{N}(0,\mathrm{I})} p_t(\mathbf{x}_t | \epsilon)$ are consistent with data $p_0(\mathbf{x})$ and noise $p_1(\mathbf{x})$ distirbution. $p_1(\mathbf{x})$ usually be chosen as gaussion distribution $\mathcal{N}(0, 1)$.

Different forward path from data to noise leads to different training object which significantly affect the performance of the model. Next we will introduce DDPM and Rectified Flow.

### C.1 DENOSING DIFFUSION PROBABILISTIC MODELS (DDPM)

In DDPM the choice for $\alpha_t$ and $\sigma_t$ is referred to as the noise schedule and the signal-to-noise-ratio (SNR) $\alpha_t^2/\sigma_t^2$ is strictly decreasing w.r.t $t$ (Kingma et al., 2021). Moreover, (Kingma et al., 2021) prove that the following stochastic differential Eq. (SDE) has same transition distribution as $p_t(\mathbf{x}_t|\mathbf{x}_0)$ for any $t \in [0, 1]$:

$$\mathrm{d}\mathbf{x}_t = f(t)\mathbf{x}_t\mathrm{d}t + g(t)\mathrm{d}\mathbf{w}_t, t \in [0, 1], \mathbf{x}_0 \sim p_0(\mathbf{x}_0), \tag{18}$$

where $\mathbf{w}_t \in \mathbb{E}^D$ is the standard Wiener process, and

$$f(t) = \frac{\mathrm{d}\log\alpha_t}{\mathrm{d}t}, \quad g^2(t) = \frac{\mathrm{d}\sigma_t^2}{\mathrm{d}t} - 2\frac{\mathrm{d}\log\alpha_t}{\mathrm{d}t}\sigma_t^2. \tag{19}$$

(Song et al., 2021) proved that the forward path in Eq. 18 has an equivalent reverse process from time 1 to 0 under some regularity conditions, starting with $p_T(\mathbf{x}_T)$

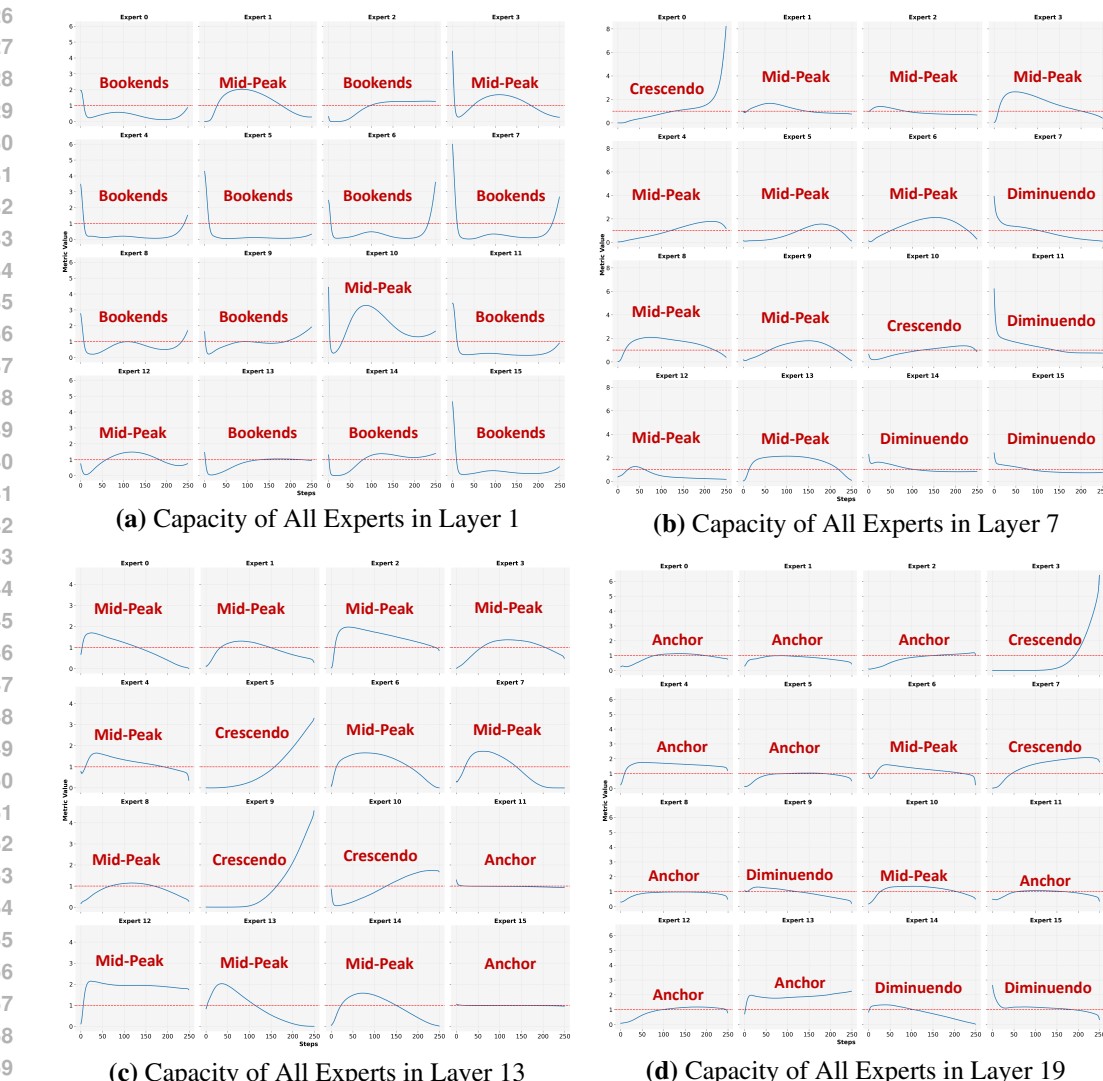

(a) Capacity of All Experts in Layer 1

(b) Capacity of All Experts in Layer 7

(c) Capacity of All Experts in Layer 13

(d) Capacity of All Experts in Layer 19

Figure 9: **Expert Dynamics across Network Layers.** Visualization of expert capacity patterns in network layers (1, 7, 13, 19). Early layers show high-amplitude fluctuations, while deeper layers exhibit increasingly stable utilization, demonstrating natural expert specialization throughout the diffusion process.

$$\mathrm{d}\mathbf{x}_t = [f(t)\mathbf{x}_t - g(t)^2 \nabla_{\mathbf{x}} \log p_t(\mathbf{x}_t)]\mathrm{d}t + g(t)\mathrm{d}\bar{\mathbf{w}}_t, \quad \mathbf{x}_T \sim p_T(\mathbf{x}_T), \tag{20}$$

where $\bar{\mathbf{w}}_t \in \mathbb{E}^D$ is the standard Wiener process. We can esitimate the score term $\nabla_{\mathbf{x}} \log p_t(\mathbf{x}_t)$ as each time $t$ to iteratively solve the reverse process, then get the gernerated target. DPMs train a neural network $\epsilon_\theta(\mathbf{x}, t)$ parameterized by $\theta$ to esitimated the scaled score function $-\sigma \nabla_{\mathbf{x}} \log p_t(\mathbf{x}_t)$. To optimize $\epsilon_\theta$, we minimize the following objective (Ho et al., 2020; Song et al., 2021; Ma et al., 2024)

$$\mathcal{L}_{\mathrm{DDPM}}(\theta) = \mathbb{E}_{t,p_0(\mathbf{x}_0),p(\mathbf{x}_t|\mathbf{x}_0)} \left[ \lambda(t) \| \epsilon_\theta(\mathbf{x}_t, t) + \sigma_t \nabla_{\mathbf{x}} \log p_t(\mathbf{x}_t) \|_2^2 \right], \tag{21}$$

where $\lambda_t$ is a time-dependent coefficent. $\epsilon_\theta(\mathbf{x}_t, t)$ can be interpreted as predicting the Gaussian noise added to $\mathbf{x}_t$, thus it is commonly referred to as a noise prediction model. Consequently, the diffusion model is known as a denoising diffusion probabilistic model. Substitute score term in equation 20 with $-\epsilon_\theta(\mathbf{x}_t, t)/\sigma_t$, we can solve the reverse process and generate samples from DPMs

with numerical solvers. To further accelerate the sampling process, (Song et al., 2021) proved that the equvivalent probability flow ODE is

$$\frac{\mathrm{d}\mathbf{x}_t}{\mathrm{d}t} = \mathbf{v}_\theta(\mathbf{x}_t, t) := f(t)\mathbf{x}_t + \frac{g^2(t)}{2\sigma_t}\epsilon_\theta(\mathbf{x}_t, t), \quad \mathbf{x}_1 \sim \mathcal{N}(0, \mathbf{I}). \tag{22}$$

Thus samples can be also generated by solving the ODE from time 1 to 0.

## C.2   RECTIFIED FLOW MODELS (FLOW)

Recified flow models (Liu et al., 2022b; Albergo & Vanden-Eijnden, 2023; Lipman et al., 2023) connects data $\mathbf{x}_0$ and noise $\epsilon$ on a straight line as follows

$$\mathbf{x}_t = (1 - t)\mathbf{x}_0 + t\epsilon, \quad t \in [0, 1]. \tag{23}$$

To precisely express the relationship between $\mathbf{x}_t$, $\mathbf{x}_0$, and $\epsilon$, we first construct a time-dependent vector field $u : [0, 1] \times \mathbb{R}^D \to \mathbb{R}^D$. This vector field $u_t$ can be used to construct a time-dependent diffeomorphic map, known as a flow $\phi : [0, 1] \times \mathbb{R}^D \to \mathbb{R}^D$, through the following ODE:

$$\frac{\mathrm{d}}{\mathrm{d}t}\phi_t(\mathbf{x}_0) = u_t(\phi_t(\mathbf{x}_0)) \tag{24}$$

$$\phi_0(\mathbf{x}_0) = \mathbf{x}_0. \tag{25}$$

The vector field $u_t$ can be modeled as a neural network $\mathbf{v}_\theta$ (Chen et al., 2018) which leads to a deep parametric model of the flow $\phi_t$, called a *Continuous Normalizing Flow* (CNF). We can using *conditional flow matching* (CFM) technique (Lipman et al., 2022) to training a CNF. Now we can define the flow we need as follows:

$$\psi_t(\cdot|\epsilon) : \mathbf{x}_0 \mapsto \alpha_t\mathbf{x}_0 + \sigma_t\epsilon. \tag{26}$$

The corresponding velocity vector field of the flow $\psi_t$ can be represented as:

$$u_t(\psi_t(\cdot|\epsilon)|\epsilon) = \frac{\mathrm{d}}{\mathrm{d}t}\psi_t(\mathbf{x}_0|\epsilon) = \dot{\alpha}_t\mathbf{x}_0 + \dot{\sigma}_t\epsilon = \epsilon - \mathbf{x}_0. \tag{27}$$

Using conditional flow matching technique, $\mathbf{v}(\mathbf{x}_t, t)$ in Eq. equation 22 can be modeled as a neural network $\mathbf{v}_\theta(\mathbf{x}_t, t)$ by minimziing the following objective

$$\mathcal{L}_{\mathrm{Flow}}(\theta) = \mathbb{E}_{t, p_0(\mathbf{x}_0), p_1(\epsilon)}\|\mathbf{v}_\theta(\mathbf{x}_t, t) - \frac{\mathrm{d}}{\mathrm{d}t}\psi_t(\mathbf{x}_0|\epsilon)\|_2^2 \tag{28}$$

$$= \mathbb{E}_{t, p_0(\mathbf{x}_0), p(\epsilon)}\|\mathbf{v}_\theta(\mathbf{x}_t, t) - (\dot{\alpha}_t\mathbf{x}_0 + \dot{\sigma}_t\epsilon)\|_2^2 \tag{29}$$

$$= \mathbb{E}_{t, p_0(\mathbf{x}_0), p(\epsilon)}\|\mathbf{v}_\theta(\mathbf{x}_t, t) - (\epsilon - \mathbf{x}_0)\|_2^2. \tag{30}$$

Samples can be generated by sovling the probability flow ODE below with learned velocity using numerical sovler like Euler, Heun, Runge-Kutta method.

$$\mathrm{d}\mathbf{x}_t = \mathbf{v}_\theta(\mathbf{x}_t, t)\mathrm{d}t, \quad \mathbf{x}_1 = \epsilon \sim \mathcal{N}(0, 1). \tag{31}$$

## C.3   RELATIONSHIP DDPM AND FLOW

There exists a straightforward connection between $\mathbf{v}_\theta(\mathbf{x}_t, t)$ and the score term $-\sigma_t\nabla_\mathbf{x}\log p_t(\mathbf{x}_t)$ can be derived as follows

$$\mathbf{v}_\theta(\mathbf{x}_t, t) = f(t)\mathbf{x}_t + \frac{g^2(t)}{2\sigma_t}\epsilon_\theta(\mathbf{x}_t, t) \tag{32}$$

$$\approx \frac{\dot{\alpha}_t}{\alpha_t}\mathbf{x}_t + \left(\dot{\sigma}_t - \frac{\dot{\alpha}_t}{\alpha_t}\right)(-\sigma_t\nabla_\mathbf{x}\log p_t(\mathbf{x}_t)). \tag{33}$$

Let $\zeta_t = \dot{\sigma}_t - \frac{\dot{\alpha}_t}{\alpha_t}$, and we have $\epsilon_\theta(\mathbf{x}_t, t) \approx -\sigma_t \nabla_{\mathbf{x}} \log p_t(\mathbf{x}_t)$, then we can get $\mathbf{v}_\theta(\mathbf{x}_t, t) = \frac{\dot{\alpha}_t}{\alpha_t} \mathbf{x}_t + \zeta_t \epsilon_\theta(\mathbf{x}_t, t)$

By plugging equation 33 into the loss $\mathcal{L}_{\text{Flow}}$ in Eq. equation 30 we have:

$$\mathcal{L}_{\text{Flow}}(\theta) = \mathbb{E}_{t, p_0(\mathbf{x}_0), p_1(\epsilon)} \|\mathbf{v}_\theta(\mathbf{x}_t, t) - (\dot{\alpha}_t \mathbf{x}_0 + \dot{\sigma}_t \epsilon)\|_2^2 \tag{34}$$

$$= \mathbb{E}_{t, p_0(\mathbf{x}_0), p(\epsilon)} \|\frac{\dot{\alpha}_t}{\alpha_t} \mathbf{x}_t + \zeta_t \epsilon_\theta(\mathbf{x}_t, t) - \dot{\sigma}_t \epsilon\|_2^2 \tag{35}$$

$$= \mathbb{E}_{t, p_0(\mathbf{x}_0), p(\epsilon)} \|\zeta_t \epsilon_\theta(\mathbf{x}_t, t) - \zeta_t \epsilon\|_2^2 \tag{36}$$

$$= \mathbb{E}_{t, p_0(\mathbf{x}_0), p(\epsilon)} \left[ \zeta_t^2 \|\epsilon_\theta(\mathbf{x}_t, t) - \epsilon\|_2^2 \right]. \tag{37}$$

Considering Eq. equation 17, we have $\mathbf{x}_t \sim \mathcal{N}(\alpha_t \mathbf{x}_0, \sigma_t \mathbf{I})$ and $\nabla_{\mathbf{x}} \log p(\mathbf{x}_t) = \sigma_t^{-1}(\mathbf{x}_t - \alpha_t \mathbf{x}_0) = \sigma_t^{-1}(\sigma_t \epsilon) = \epsilon$. Then, we can get the equivilant loss function as below:

$$\mathcal{L}_{\text{Flow}}(\theta) = \mathbb{E}_{t, p_0(x_0), p(\mathbf{x}_t | \mathbf{x}_0)} \left[ \zeta_t^2 \|\epsilon_\theta(\mathbf{x}_t, t) + \sigma_t \nabla_{\mathbf{x}} \log p_t(\mathbf{x}_t)\|_2^2 \right]. \tag{38}$$

Recall that $\mathcal{L}_{\text{DDPM}}(\theta) = \mathbb{E}_{t, p_0(\mathbf{x}_0), p(\mathbf{x}_t | \mathbf{x}_0)} \left[ \lambda(t) \|\epsilon_\theta(\mathbf{x}_t, t) + \sigma_t \nabla_{\mathbf{x}} \log p_t(\mathbf{x}_t)\|_2^2 \right].$

We can find that $\mathcal{L}_{\text{DDPM}}$ and $\mathcal{L}_{\text{Flow}}$ have the same form, with the only difference being their time-dependent weighting functions, which lead to different trajectories and properties.

## D MORE IMPLEMENTATION DETAILS

### D.1 TRAINING SETUP.

We train class-conditional DiffMoE and baseline models at 256x256 image resolution on the ImageNet dataset (Russakovsky et al., 2015) a highly-competitive generative benchmark, which contains 1281167 training images. We use horizontal flips as the only data augmentation. We train all models with AdamW (Loshchilov & Hutter, 2017). We use a constant learning rate of $1 \times 10^{-4}$, no weight decay and a fixed global batch size of 256, following (Peebles & Xie, 2023b). We also maintain an exponential moving average(EMA) of DiffMoE and baseline model weights over training with a decay of 0.9999. All results reported use the EMA mode. For most experiments, we utilized 4 NVIDIA H800 GPUs during training. To achieve state-of-the-art results, we extended the training process with 8 NVIDIA H800 GPUs for improved efficiency. For text-to-image experiments, we train all baseline models Dense-DiT, TC-DiT, EC-DiT, and our DiffMoE on a dataset of 25M 256×256 text-image pairs (mixed from LAION (Schuhmann et al., 2021) and JourneyDB (Pan et al., 2023)) for 576 H800 GPU hours.

### D.2 IMPLEMENTATION ALGORITHMS.

We provide a detailed illustration of the DiffMoE layer during training and inference in Algorithm 3 and 4, respectively. We also implemented the same EC-DiT layer in Algorithm 1 as (Sun et al., 2024). We implemented TC-DiT in layer in Algorithm 2 similar to (Dai et al., 2024; Fei et al., 2024)

## E CALCULATION OF AVERAGE ACTIVATED PARAMETERS AND AVERAGE CAPACITY

### E.1 COMPUTING AVERAGE INFERENCE CAPACITY

To compute the global average capacity ($C_{\text{infer}}^{\text{avg}}$), we analyze 50K samples across all experts and sampling steps. For a quick approximation, sampling 1K samples is sufficient due to DiffMoE's stable performance characteristics.

### E.2 ESTIMATING AVERAGE ACTIVATED PARAMETERS USING AVERAGE INFERENCE CAPACITY

We will introduce the relationship between average activated parameters and average capacity in detail. Let #Module denote the number of parameters a certain Module, $N_E$ denote the number of experts. Under approximate conditions, for DiT (Peebles & Xie, 2023a) models, we have

In our model, we adopt an interlaced arrangement of the Mixture of Experts (MoE) layer and the dense layer. Here, #FFN represents the total number of parameters of all Feed - Forward Neural Network (FFN) layers. Given that the total number of layers is $N$, the calculation of the average activated parameters is as follows:

The contribution of FFN layers to the average activated parameters can be derived step - by - step. First, consider the proportion of active FFN layers. Since MoE and dense layers are interlaced, the number of active FFN layers is approximately $N/2$.

The formula for the average activated parameters of FFN layers is:

$$\text{\# Average Activated Parameters of FFN layers}$$
$$\approx \frac{N/2}{N/2 + N/2 \times N_E} \times \text{\#FFN} + \frac{N/2 \times C_{\text{infer}}^{\text{avg}}}{N/2 + N/2 \times N_E} \times \text{\#FFN} = \left( \frac{1 + C_{\text{infer}}^{\text{avg}}}{1 + N_E} \right) \times \text{\#FFN}$$

Combining with the contributions from other components (Attention, AdaLN, and other modules), the overall average activated parameters of the model can be approximated as:

$$\text{\# Average Activated Parameters} \approx \left( \frac{1 + C_{\text{infer}}^{\text{avg}}}{1 + N_E} \right) \text{\#FFN} + \text{\#Attention} + \text{\#AdaLN} + \text{\#Other Modules} \tag{39}$$

Table 10 displays the parameters and their corresponding percentages of the main modules of large-size DiffMoE.

## F FID SENSIBILITY AND ABLATION STUDY

FID scores are sensitive to implementation details, necessitating careful ablation studies to understand the differences between various implementations. Through these studies, we aim to provide the academic community with clearer insights for fair comparisons between diffusion models.

The observed FID degradation at higher CFG scales is well-documented (Ma et al., 2024), primarily due to ImageNet's diverse image quality distribution. When generating high-quality samples, the deviation from ImageNet's mixed-quality dataset can lead to increased FID scores, despite improved visual quality.

For FID calculation, we follow the implementations from SiT (Ma et al., 2024) and DiT (Peebles & Xie, 2023a). Results marked with $^\dagger$ are directly quoted from (Ma et al., 2024) (DDPM) and (Peebles & Xie, 2023a) (Flow). For results marked with $^*$, we reproduce the experiments using officially released checkpoints under identical evaluation conditions.

### F.1 VAE DECODER ABLATIONS

Following (Peebles & Xie, 2023a), throughout our experiments, we employed pre-trained VAE models. Specifically, we utilized fine-tuned versions (ft-MSE and ft-EMA) of the original LDM "f8" model, where only the decoder weights were fine-tuned. For the analysis presented in Experiments section 3 and Tables 12 and 11, we tracked metrics using the ft-MSE decoder, while the final metrics reported in Table 2 was obtained using the ft-EMA decoder. In this section, we examine the impact of two distinct VAE decoders for our experiments - the two fine-tuned variants employed in Stable Diffusion. Since all models share identical encoders, we can interchange decoders without necessitating diffusion model retraining. As demonstrated in Table 13, the DiffMoE model maintains its superior performance over existing diffusion models.

Table 17: **Batch Size Ablation Study:** FID scores under different batch sizes using fine-tuned EMA VAE decoder and Heun sampler. **Bold** indicates best performance in its cell.

| Model | Training Steps | CFG | Batch Size* | FID50K ↓ |
|---|---|---|---|---|
| DiffMoE-L-E8-Flow | 7000K | 1.0 | 10 | 9.60 |
| DiffMoE-L-E8-Flow | 7000K | 1.0 | 15 | 9.62 |
| DiffMoE-L-E8-Flow | 7000K | 1.0 | 32 | 9.77 |
| DiffMoE-L-E8-Flow | 7000K | 1.0 | 50 | 9.98 |
| DiffMoE-L-E8-Flow | 7000K | 1.0 | 75 | 9.90 |
| DiffMoE-L-E8-Flow | 7000K | 1.0 | 100 | 9.76 |
| DiffMoE-L-E8-Flow | 7000K | 1.0 | 125 | 9.78 |
| Dense-DiT-XL-Flow* | 7000K | 1.0 | 10 | 9.57 |
| Dense-DiT-XL-Flow* | 7000K | 1.0 | 15 | 9.47 |
| Dense-DiT-XL-Flow* | 7000K | 1.0 | 50 | 9.84 |
| Dense-DiT-XL-Flow* | 7000K | 1.0 | 75 | 9.67 |
| Dense-DiT-XL-Flow* | 7000K | 1.0 | 100 | 9.65 |
| Dense-DiT-XL-Flow* | 7000K | 1.0 | 125 | 9.64 |
| DiffMoE-L-E8-Flow | 7000K | 1.5 | 10 | 2.19 |
| DiffMoE-L-E8-Flow | 7000K | 1.5 | 32 | 2.16 |
| DiffMoE-L-E8-Flow | 7000K | 1.5 | 50 | 2.16 |
| DiffMoE-L-E8-Flow | 7000K | 1.5 | 75 | 2.13 |
| DiffMoE-L-E8-Flow | 7000K | 1.5 | 100 | 2.17 |
| DiffMoE-L-E8-Flow | 7000K | 1.5 | 125 | 2.18 |
| Dense-DiT-XL-Flow* | 7000K | 1.5 | 10 | 2.23 |
| Dense-DiT-XL-Flow* | 7000K | 1.5 | 15 | 2.23 |
| Dense-DiT-XL-Flow* | 7000K | 1.5 | 50 | 2.21 |
| Dense-DiT-XL-Flow* | 7000K | 1.5 | 75 | 2.19 |
| Dense-DiT-XL-Flow* | 7000K | 1.5 | 100 | 2.22 |
| Dense-DiT-XL-Flow* | 7000K | 1.5 | 125 | 2.21 |
| DiffMoE-L-E8-DDPM | 7000K | 1.5 | 50 | 2.30 |
| DiffMoE-L-E8-DDPM | 7000K | 1.5 | 75 | 2.33 |
| DiffMoE-L-E8-DDPM | 7000K | 1.5 | 100 | 2.32 |
| DiffMoE-L-E8-DDPM | 7000K | 1.5 | 125 | 2.32 |

Table 18: **CFG Scale Ablation Study:** FID scores across different CFG scales using fine-tuned EMA VAE decoder. **Bold** indicates best performance in its cell.

| Model | Training Steps | Sampler | CFG | Batch Size | FID50K ↓ |
|---|---|---|---|---|---|
| DiffMoE-L-E8-Flow | 4900K | Heun | 1.0 | 125 | 9.21 |
| Dense-DiT-XL-Flow* | 7000K | Heun | 1.0 | 125 | 9.64 |
| DiffMoE-L-E8-Flow | 7000K | Heun | 1.0 | 125 | 9.78 |
| DiffMoE-L-E8-Flow | 4900K | Heun | 1.43 | 125 | 2.14 |
| Dense-DiT-XL-Flow* | 7000K | Heun | 1.43 | 125 | 2.08 |
| DiffMoE-L-E8-Flow | 7000K | Heun | 1.43 | 125 | 2.13 |
| DiffMoE-L-E8-Flow | 4900K | Heun | 1.5 | 125 | 2.28 |
| Dense-DiT-XL-Flow* | 7000K | Heun | 1.5 | 125 | 2.21 |
| DiffMoE-L-E8-Flow | 7000K | Heun | 1.5 | 125 | 2.18 |
| DiffMoE-L-E8-DDPM | 6500K | DDPM | 1.0 | 125 | 9.39 |
| Dense-DiT-XL-DDPM* | 7000K | DDPM | 1.0 | 125 | 9.63 |
| DiffMoE-L-E8-DDPM | 7000K | DDPM | 1.0 | 125 | 9.17 |
| DiffMoE-L-E8-DDPM | 6500K | DDPM | 1.5 | 125 | 2.27 |
| Dense-DiT-XL-DDPM* | 7000K | DDPM | 1.5 | 125 | 2.32 |
| DiffMoE-L-E8-DDPM | 7000K | DDPM | 1.5 | 125 | 2.32 |

Table 19: **Flow ODE Sampler Ablation Study:** FID scores across different ODE samplers with CFG scale 1.0 and fine-tuned EMA VAE decoder. **Bold** indicates best performance in its cell.

| Model | Training Steps | Sampler | Batch Size | FID50K ↓ |
|---|---|---|---|---|
| DiffMoE-L-E8-Flow | 4900K | Euler | 125 | 9.37 |
| DiffMoE-L-E8-Flow | 4900K | Heun | 125 | 9.21 |
| DiffMoE-L-E8-Flow | 4900K | Euler | 250 | 9.39 |
| DiffMoE-L-E8-Flow | 4900K | Dopri5 | 250 | 9.06 |
| DiffMoE-L-E8-Flow | 7000K | Euler | 125 | 9.86 |
| DiffMoE-L-E8-Flow | 7000K | Heun | 125 | 9.78 |
| DiffMoE-L-E8-Flow | 7000K | Euler | 250 | 9.94 |
| DiffMoE-L-E8-Flow | 7000K | Dopri5 | 250 | 9.56 |

## F.2 FLOW ODE-SAMPLER ABLATIONS

Higher-order ODE samplers generally achieve better FID scores. As shown in Table 19, the black-box dopri5 sampler outperforms heun (NFE=250), which in turn surpasses euler (NFE). For fair comparison with baseline models, we employ the euler sampler in flow-based experiments. However, to benchmark against SiT-XL (Dense-DiT-XL-Flow) (Ma et al., 2024), we use the heun sampler to achieve SOTA results.

## F.3 CLASSIFIER-FREE-GUIDANCE ABLATIONS

We evaluate different models with varying classifier-free guidance (CFG) scales in Table 18, and discover that the CFG scale of 1.5 adopted in DiT (Peebles & Xie, 2023a) and SiT (Ma et al., 2024) studies may not be universally optimal. Our analysis reveals the best CFG scale approximates to 1.43 through comprehensive comparisons. However, different models exhibit distinct characteristics that lead to varying optimal CFG scales, suggesting that fixing a uniform scale across all models could introduce evaluation bias. To ensure relatively fair comparisons while maintaining consistency with established practices, we ultimately adopt CFG 1.5 as the default setting in our experiments. This decision aligns with the well-documented trade-off in diffusion models: higher CFG scales (e.g., 4.0) typically enhance image fidelity at the cost of increased FID scores, while lower scales (e.g., 1.5) yield better FID metrics despite reduced perceptual quality. This phenomenon primarily stems from FID's sensitivity to distributional coverage - higher guidance scales tend to produce samples with reduced diversity that more closely match the training distribution statistics, paradoxically resulting in worse FID scores despite improved individual sample quality.

## F.4 BATCH SIZES ABLATIONS

The batch sizes ablation study reveals critical insights into the interplay between batch size and classifier-free guidance (CFG) scales for the DiffMoE-L-E8-Flow model. At CFG=1.0, FID scores remain elevated (9.60–9.98), with smaller batch sizes (*e.g.*, `bs=10`) marginally outperforming larger

---

**Algorithm 1** EC - DiT Layer

---

**Require:** $x$ (input tensor)

   $B$ (batch size), $S$ (sequence length), $d$ (hidden dim), $W_r$ (routing weights), experts (list of expert FFNs)

   $E$ (number of experts), $C$ (expert capacity)

1: **Step 1: Compute Token - Expert Affinity Matrix**
2: $\text{logits} \leftarrow \text{einsum}('bsd, de \rightarrow bse', x, W_r)$
3: $\text{scores} \leftarrow \text{softmax}(\text{logits}, \dim = -1).\text{permute}(-1, -2)$
4: **Step 2: Select top - k tokens for each expert**
5: $\text{gating, index} \leftarrow \text{top\_k}(\text{scores}, k = C, \dim = -1)$
6: $\text{dispatch} \leftarrow \text{one\_hot}(\text{index}, \text{num\_classes} = S)$
7: **Step 3: Process tokens through experts and combine**
8: $x_{in} \leftarrow \text{einsum}('becs, bsd \rightarrow becd', \text{dispatch}, x)$
9: $x_e \leftarrow [\text{experts}[e](x_{in}[:, e]) \text{ for } e \text{ in range}(E)]$
10: $x_e \leftarrow \text{stack}(x_e, \dim = 1)$
11: $x_{out} \leftarrow \text{einsum}('becs, bec, becd \rightarrow bsd', \text{dispatch}, \text{gating}, x_e)$

**Ensure:** $x_{out}$

---

configurations, exhibiting a U-shaped trend. However, elevating CFG to 1.5 drastically reduces FID to 2.13–2.19, achieving optimal performance at `bs=75` (**2.13**), while demonstrating remarkable robustness to batch size variations ($\Delta$=0.06 vs. $\Delta$=0.38 at CFG=1.0).

### F.5 CONCLUSION: A LITTLE THOUGHT ABOUT FID

While Fréchet Inception Distance (FID) is widely adopted for evaluating generative models, particularly on ImageNet, it exhibits several notable limitations. Our analysis reveals counterintuitive behaviors, especially when evaluating models with classifier-free guidance (CFG). For example, higher CFG scales typically enhance perceptual quality but paradoxically result in worse FID scores, despite producing visually superior images. This discrepancy stems from FID's fundamental mechanism: it measures statistical similarities between generated and real distributions in the Inception network's feature space, often failing to capture perceptual quality and fine-grained details. Moreover, FID scores are susceptible to various implementation factors, including choice of ODE samplers, hardware configurations, random seeds, and sample size for estimation. These sensitivities can impact reproducibility and comparison across different studies. Furthermore, FID's focus on distributional overlap overlooks critical aspects such as mode collapse and overfitting, as it does not explicitly evaluate sample diversity or novelty. These limitations underscore the pressing need for more robust and comprehensive metrics that can better reflect the true modeling capabilities of generative models. We advocate for developing new evaluation frameworks that combine precision-recall curves, perceptual quality metrics, and human evaluation studies, which would provide a more reliable assessment of generative model performance.

## G VISUAL GENERATION RESULTS

### G.1 CLASS-CONDITIONAL IMAGE GENERATION

To demonstrate the generation capabilities of our model, we showcase diverse images sampled from DiffMoE-L-E8-Flow and DiffMoE-L-E8-DDPM, conditioned on ImageNet class labels. These visualizations illustrate the model's ability to generate high-quality, class-specific images. See Figure 10 and 11.

### G.2 TEXT-CONDITIONAL IMAGE GENERATION

We present a collection of images generated by our DiffMoE-T2I-Flow model using various text prompts as conditioning inputs. These examples demonstrate the model's versatility in translating textual descriptions into corresponding visual representations. See Figure 12.

---

**Algorithm 2** TC - DiT layer

---

**Require:** $x$ (input tensor), $W_r$ (routing weights), experts (list of expert FFNs)
$\qquad$ $B$ (batch size), $S$ (sequence length), $d$ (hidden dim), $K$ (experts per token)
1: **Step 1: Save original input shape**
2: orig_shape $\leftarrow$ shape$(x)$
3: **Step 2: Compute Token - Expert Affinity Matrix**
4: logits $\leftarrow$ einsum('$bsd, de \rightarrow bse'$, $x, W_r$)
5: scores $\leftarrow$ softmax(logits, dim $= -1$)
6: **Step 3: Select top - k tokens for each expert**
7: gating, index $\leftarrow$ top_k(scores, $k = K$, dim $= -1$)
8: **Step 4: Flatten $x$ and top - k indices**
9: $x \leftarrow$ view$(x, (-1, x.shape[-1]))$
10: flat_topk_idx $\leftarrow$ view(topk_idx, $(-1)$)
11: **Step 5: Process tokens through experts**
12: $x \leftarrow$ repeat_interleave$(x, K, \text{dim} = 0)$
13: $y \leftarrow$ empty_like$(x)$
14: **for** $i \leftarrow 1$ to len(experts) **do**
15: $\quad$ $y$[flat_topk_idx $== i$] $\leftarrow$ expert$_i$($x$[flat_topk_idx $== i$])
16: **end for**
17: $y \leftarrow$ sum(view$(y, (*\text{gating}.shape, -1)) \cdot \text{gating}.unsqueeze(-1)$, dim $= 1$)
18: $y \leftarrow$ view$(y, \text{orig\_shape})$
**Ensure:** $y$

---

**Algorithm 3** DiffMoE layer (Training)

---

**Require:** $x$ (input tensor)
$\qquad$ $B$ (batch size), $S$ (flattened sequence length), $D$ (hidden dim), $N$ (number of experts)
$\qquad$ $W_r$ (routing weights), experts (list of expert FFNs), $C$ (expert capacity)
1: **Step 1: Batch-level token pool and compute capacity prediction**
2: $x \leftarrow$ view$(x, (-1, D))$
3: $S \leftarrow$ shape$(x)[0]$
4: capacity_pred $\leftarrow$ capacity_predictor(detach$(x)$)
5: $C_{\text{train}} \leftarrow$ int$((S/N) \times C)$
6: **Step 2: Compute token - expert affinity scores**
7: logits $\leftarrow$ einsum('$sd, de \rightarrow se'$, $x, W_r$)
8: scores $\leftarrow$ softmax(logits, dim $= -1$).permute$(-1, -2)$
9: gating, index $\leftarrow$ top_k(scores, $k = C_{\text{train}}$, dim $= -1$, sorted $=$ False)
10: **Step 3: Process tokens through experts**
11: $y \leftarrow$ zeros_like$(x)$, ones $\leftarrow$ zeros$(N, S)$
12: **for** $i \leftarrow 1$ to $N$ **do**
13: $\quad$ $y$[index$[i], :$] $\leftarrow y$[index$[i], :$] $+$ gating$[i]$.unsqueeze$(-1) \times$ expert$_i$($x$[index$[i], :$])
14: $\quad$ ones$[i]$[index$[i]$] $\leftarrow 1$
15: **end for**
16: **Step 4: Update capacity threshold**
17: update_threshold(capacity_pred)
18: **Step 5: Reshape output**
19: $x_{\text{out}} \leftarrow$ view$(y, (B, s, D))$
**Ensure:** $x_{\text{out}}$, ones, capacity_pred

---

# H STATEMENT ON LLM ASSISTANCE

Portions of this manuscript were refined for clarity and readability using Claude and DeepSeek. The authors remain solely responsible for the technical content and conclusions presented in this work.

**Algorithm 4** DiffMoE layer (Inference)

---

**Require:** $x$ (input tensor)

$\quad\quad\quad$ $B$ (batch size), $S$ (flattened sequence length), $D$ (hidden dim), $N$ (number of experts)

$\quad\quad\quad$ $W_r$ (routing weights), experts (list of expert FFNs), $C$ (expert capacity)

$\quad\quad\quad$ threshold (expert threshold)

1: **Step 1: Reshape input and compute capacity prediction**

2: $x \leftarrow \text{view}(x, (-1, D))$

3: $S \leftarrow \text{shape}(x)[0]$

4: $\text{capacity\_pred} \leftarrow \text{sigmoid}(\text{capacity\_predictor}(\text{detach}(x)))$

5: **Step 2: Compute token - expert affinity scores**

6: $\text{logits} \leftarrow \text{einsum}(`sd, de \rightarrow se\text{'}, x, W_r)$

7: $\text{scores} \leftarrow \text{softmax}(\text{logits}, \dim = -1).\text{permute}(-1, -2)$

8: **Step 3: Process tokens through experts**

9: $y \leftarrow \text{zeros\_like}(x)$

10: **for** $i \leftarrow 1$ to $N$ **do**

11: $\quad\quad k_{\text{pred}} \leftarrow \text{sum}(\text{where}(\text{capacity\_pred}[:, i] > \text{threshold}[i], 1, 0))$

12: $\quad\quad \text{gating}, \text{index} \leftarrow \text{top\_k}(\text{scores}[i], k = k_{\text{pred}}, \dim = -1, \text{sorted} = \text{False})$

13: $\quad\quad y[\text{index}, :] \leftarrow y[\text{index}, :] + \text{gating}.\text{unsqueeze}(-1) \times \text{expert}_i(x[\text{index}, :])$

14: **end for**

15: **Step 4: Reshape output**

16: $x_{\text{out}} \leftarrow \text{view}(y, (B, s, D))$

**Ensure:** $x_{\text{out}}$

---

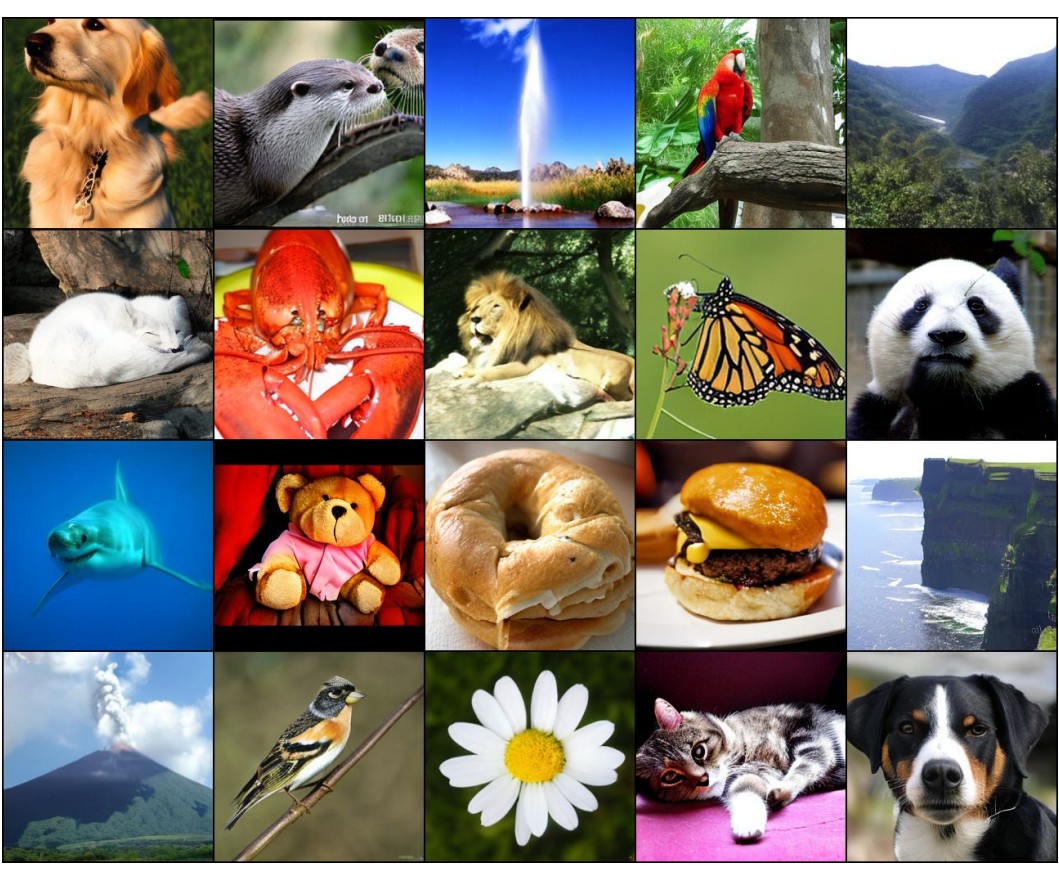

Figure 10: **Class-conditional Generation. Uncurated** $256 \times 256$ **DiffMoE-L-E8-Flow** samples CFG scale = 4.0.

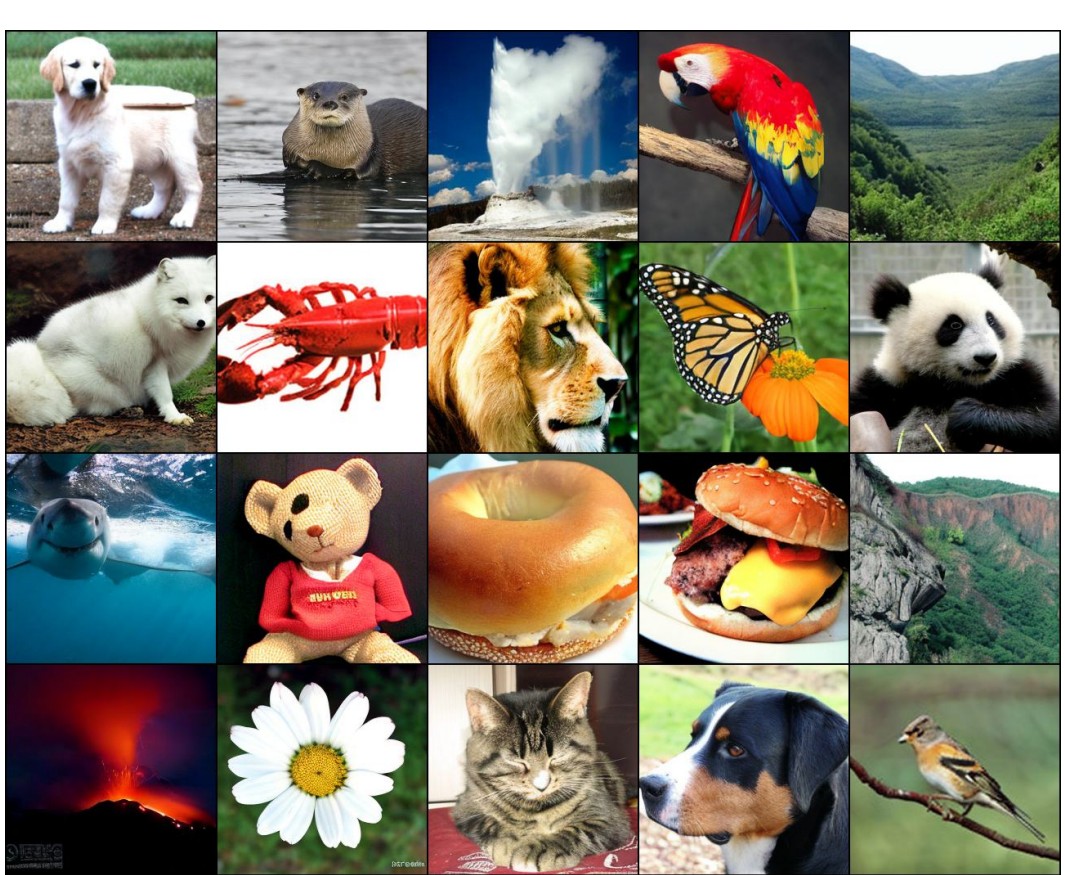

Figure 11: **Class-conditional Generation. Uncurated** 256×256 **DiffMoE-L-E8-DDPM** samples CFG scale = 4.0.

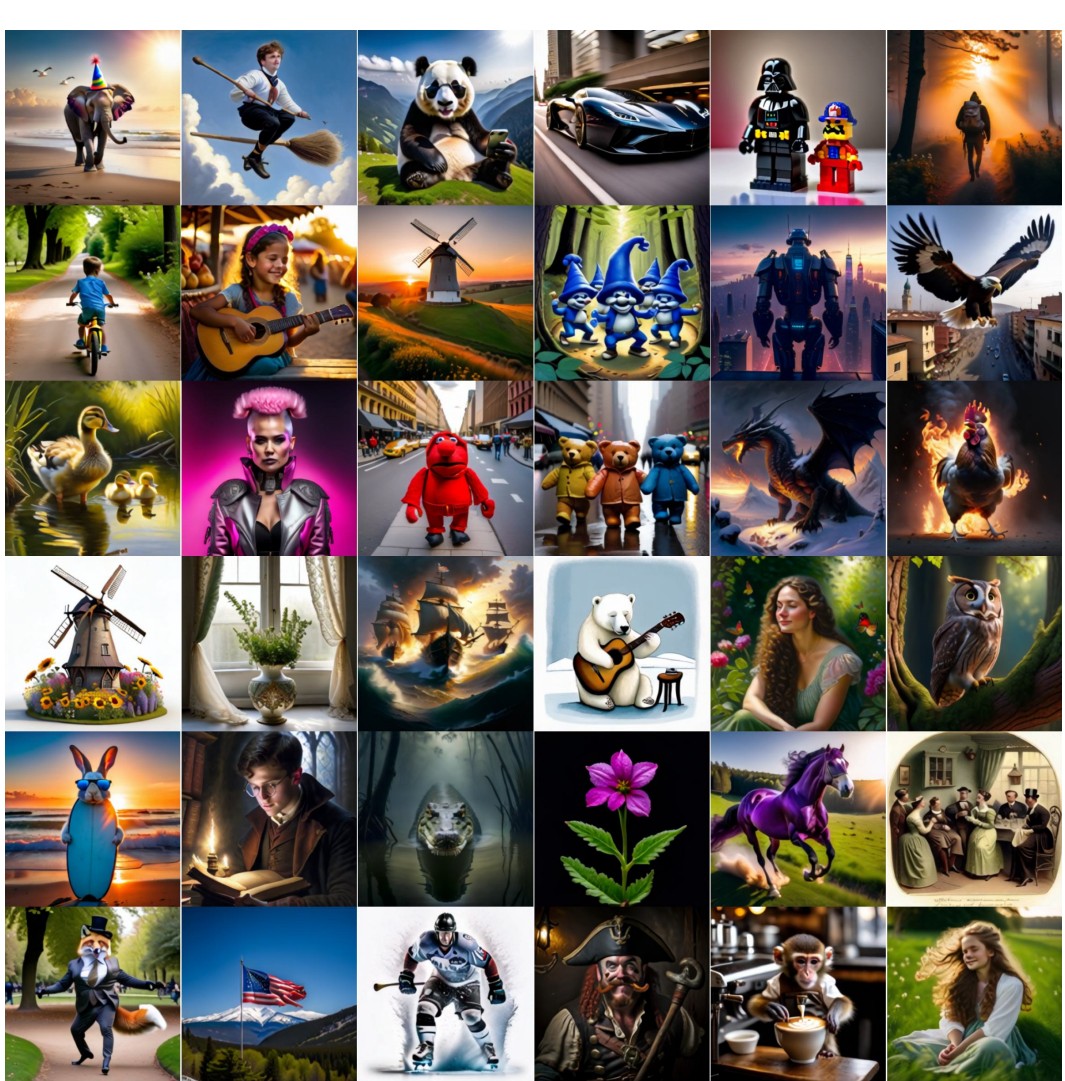

Figure 12: **Text-to-Image Generation. Uncurated** 256×256 Images generated by **DiffMoE-E16-T2I-Flow**

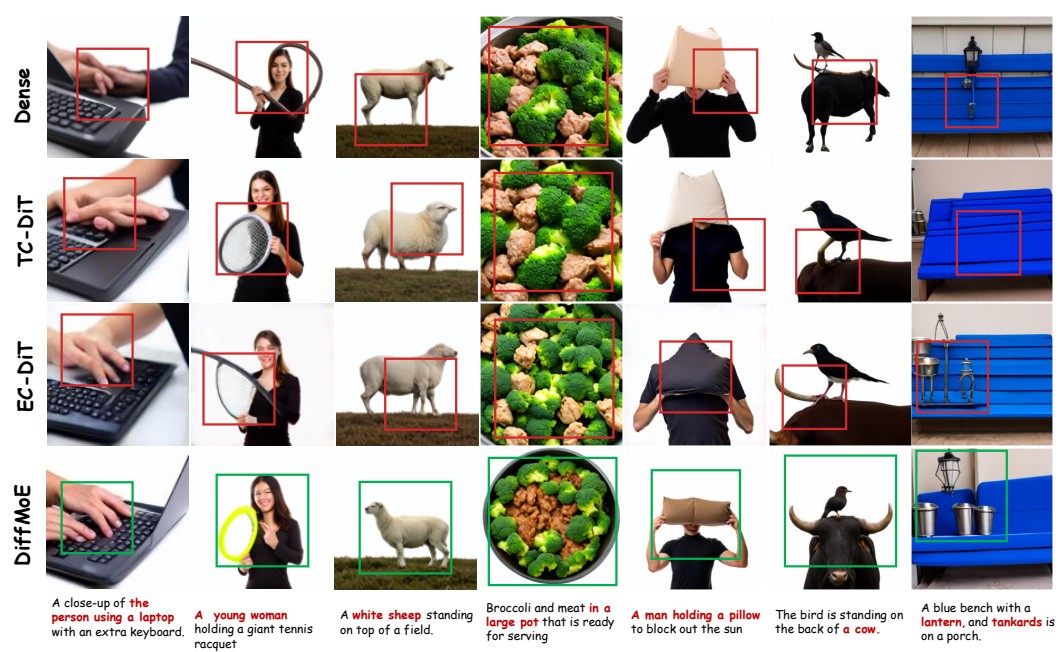

Figure 13: **Visual comparisons** among DiffMoE, Dense-DiT, TC-DiT, and EC-DiT. Across all examples shown, DiffMoE shows consistently stronger text–image alignment and more accurate visual details.

