# OpenReview forum: "DiffMoE: Dynamic Token Selection for Scalable Diffusion Transformers"
_ICLR.cc/2026/Conference — Submitted to ICLR 2026_

### Official Review · Reviewer_jjxW · 2025-10-28

**Soundness:** 4
**Presentation:** 3
**Contribution:** 3
**Rating:** 6
**Confidence:** 3

**Summary:**

This paper introduces DiffMoE, a novel method for incorporating Mixture of Experts into the training of Diffusion Transformers. The key innovation is the creation of a batch-level global token pool, achieved by flattening all tokens within a batch. This allows experts to be trained simultaneously on a diverse set of tokens from multiple images and at various noise levels. The authors show that DiffMoE achieves significant improvements over dense baselines with fewer activated parameters. The paper presents a promising and well-motivated method for training scalable and efficient MoE-based diffusion models.

**Strengths:**

- A key strength is the observation that standard MoE formulations in diffusion models prevent routers from learning from the crucial contrastive patterns that exist across different noise levels and samples. DiffMoE is designed specifically to address this limitation.
- The paper employs a fair comparison framework, carefully maintaining a fixed computational cost between the proposed method and the baselines, which strengthens the validity of the performance claims.
- The paper is well-written, clearly articulating a complex architectural change and its motivations.
- The ablation studies are thorough and effectively demonstrate the contribution of each individual component of the DiffMoE framework.

**Weaknesses:**

- The qualitative results are limited, particularly the visual comparisons against the text-to-image (T2I) baselines. More examples would help in visually assessing the performance gains.
- There is a limited ablation study on the effect of batch size. Given that the core mechanism relies on a "batch-level global token pool," batch size appears to be a critical factor that could significantly impact DiffMoE's performance.

**Questions:**

- The central premise of DiffMoE's effectiveness is that the flattened batch of tokens provides a rich, global distribution for the experts to learn from. However, in many realistic inference or fine-tuning scenarios, users are constrained to very small batch sizes (e.g., 1-4). In such a situation, the flattened "global" pool would be a poor approximation of the true training distribution. What the computational cost and performance of DiffMoE are versus the dense baselines in this practical, low-batch-size setting?

---

> ### Author Response · Authors · 2025-11-20
>
> We sincerely thank the reviewer for the positive comments on our work! We address the questions and clarify the issues accordingly as described below.
>
> > W denotes Weakness, Q denotes Questions.
>
>
>
> **W1: More visual comparisons against the text-to-image baselines**
>
> > W1: The qualitative results are limited, particularly the visual comparisons against the text-to-image (T2I) baselines. More examples would help in visually assessing the performance gains.
>
> **[Reply]**
>
> Thank you for your suggestion. **We have added additional visual comparisons with Dense-DiT, TC-DiT, and EC-DiT in Fig. 13 in updated paper. Across all examples shown, DiffMoE shows consistently stronger text–image alignment and more accurate visual details.**
>
> For instance, DiffMoE better captures fine-grained attributes such as correct hand–keyboard interactions, the proper shape of the tennis racquet, realistic sheep contours, and clear broccoli–meat in the pot. In compositional prompts (e.g., a bird standing on the back of a cow or a man holding a pillow), DiffMoE maintains accurate spatial relations, whereas baselines often produce misplaced or incomplete objects. The successful generation of images from prompts containing multiple objects (e.g., “bench, lantern, and tankards”) further demonstrates DiffMoE's ability to preserve all required elements. Since all models were trained under identical conditions (dataset, compute budget, and number of activated parameters), this provides a fair comparison. These qualitative results effectively illustrate the advantages of DiffMoE beyond what quantitative metrics and benchmarks can capture.
>
>
>
> **W2/Q1: Ablation study on the effect of batch size**
>
> > W2: There is a limited ablation study on the effect of batch size. Given that the core mechanism relies on a "batch-level global token pool," batch size appears to be a critical factor that could significantly impact DiffMoE's performance.
> >
> > Q1: The central premise of DiffMoE's effectiveness is that the flattened batch of tokens provides a rich, global distribution for the experts to learn from. However, in many realistic inference or fine-tuning scenarios, users are constrained to very small batch sizes (e.g., 1-4). In such a situation, the flattened "global" pool would be a poor approximation of the true training distribution. What the computational cost and performance of DiffMoE are versus the dense baselines in this practical, low-batch-size setting?
>
> **[Reply]**
>
> ### **1. About the Performance of DiffMoE under Small-Batch Training**
>
> Thank you for the insightful question. `DiffMoE remains effective even under small-batch training.` We conducted additional experiments with a **local batch size of 2** per GPU on 2 H800s for 400K steps (average active params: **33M**). Despite the limited batch size, **DiffMoE still benefits from inter-sample token interactions** via the global token pool.
>
> When the batch size approaches 1, the global pool naturally degenerates into **expert-choice (intra-sample) routing**, which only contains intra-sample token interaction, thus we choose 2 as the minimal effective batch size.
>
> | Model (400K, Batch Size = 2) | FID50K↓   |
> | ---------------------------- | --------- |
> | TC-DiT                       | 96.21     |
> | Dense-DiT                    | 95.19     |
> | EC-DiT                       | 93.12     |
> | **DiffMoE**                  | **89.77** |
>
> These results show that **DiffMoE consistently outperforms all baselines**, demonstrating its robustness under low-resource settings. Moreover, **when the per-GPU batch is small**, the global token pool can still be constructed via **cross-GPU communication** to preserve tokens diversity.
>
> ---
>
> ### **2. About the Performance of DiffMoE trained on Big-Batch but under Small-Batch Inference**
>
> Our capacity predictor and dynamic threshold techniques can solve the critical challenge of batch size mismatch between training and inference, ensuring stable performance. `DiffMoE maintains stable and strong performance across a wide range of batch sizes`, including batch size of 1, due to its accurate and adaptive routing mechanism. We choose our pre-trained model DiffMoE-L-E8-7000K(458M, batch pool size per GPU = 32, macro batch size = 256, 8 GPUs) to conduct the ablation study.
>
> **Key Results**:
>
> | Batch Size | FID50K↓ |
> | ---------- | ------- |
> | `1`        | `2.16`  |
> | 10         | 2.19    |
> | 75         | 2.13    |
>
> **Why It Works? Further analysis for capacity predictor to better domenstrate the high routing qulity**
>
> - The **capacity predictor** is a lightweight binary classifier trained with cross-entropy to select informative tokens.
> - Combined with a dynamic threshold, it achieves **99.53% token selection accuracy** in our experiments, ensuring high-quality routing regardless of batch size.

---

> > ### Comment · Reviewer_jjxW · 2025-11-28
> >
> > Thank you for your reply!
> >
> > Your comments address my concerns. I thus leave my positive score unchanged.

---

> ### Author Response · Authors · 2025-11-29
>
> Dear Reviewer jjxW,
>
> Thank you very much for your prompt response and for confirming that our comments have addressed your concerns. We truly appreciate you maintaining your positive score.
>
> Your thorough review and constructive feedback have been invaluable in improving the quality of our paper.
>
> Sincerely,
>
> The Authors

---

### Official Review · Reviewer_bR6L · 2025-10-29

**Soundness:** 3
**Presentation:** 2
**Contribution:** 2
**Rating:** 4
**Confidence:** 3

**Summary:**

The paper introduces DiffMoE, a novel Mixture-of-Experts (MoE) architecture that allows experts to access global token distributions via a batch-level global token pool during training, encouraging more specialized expert behavior. It further integrates a capacity predictor and dynamic thresholding mechanism to adaptively allocate computational resources based on input noise and sample complexity. Comprehensive experiments demonstrate the model’s effectiveness.

**Strengths:**

The paper introduces a novel batch-level global token pool that effectively captures the heterogeneity of diffusion processes, addressing a key limitation of prior MoE-based diffusion models. The proposed capacity predictor and dynamic thresholding mechanisms enable adaptive computation, balancing quality and efficiency. The evaluation is comprehensive, demonstrating the effectiveness of DiffMoE with significantly fewer active parameters.

**Weaknesses:**

The paper lacks an in-depth analysis of expert behavior, such as what patterns each expert learns or how tokens are distributed across experts. In addition, the batch-level global token pool may pose scalability challenges when extending to high-resolution images or video generation.

**Questions:**

1. Could the authors provide a more in-depth analysis of expert behavior, specifically how the batch-level global token pool contributes to expert specialization and growth?
2. How well does the capacity predictor generalize across different datasets or tasks? Does it require retraining for each new domain?
3. How does DiffMoE perform on high-resolution tasks, and are there any scalability limitations?

---

> ### Author Response · Authors · 2025-11-20
>
> We sincerely thank the reviewer for the positive comments on our work! We address the questions and clarify the issues accordingly as described below.
>
> > W denotes Weakness, Q denotes Questions.
>
> **W1/Q1: About more in-depth analysis of expert bahavior**
>
> > W1: The paper lacks an in-depth analysis of expert behavior, such as what patterns each expert learns or how tokens are distributed across experts.
> >
> > Q1: Could the authors provide a more in-depth analysis of expert behavior, specifically how the batch-level global token pool contributes to expert specialization and growth?
>
> **[Reply]**
>
> Thank you for the insightful question. In our revision, we added a clearer explanation of how expert specialization emerges, focusing on the role of the **batch-level global token pool**. **The batch-level global token pool enhances cross-sample token interaction and drives clearer, more stable expert specialization during training.**
>
> The batch-level token pool works by **collecting all tokens from every sample in the batch** and performing routing on this global set instead of per-sample. This design allows tokens from different timesteps and prompt (class) conditions, which naturally encourages experts to specialize. For example, some experts become more aligned with early noisy timesteps, while others specialize in semantic or fine-detail regions.
>
> ### **1. Expert Specialization Analysis**
>
> Our core finding is that expert specialization in DiffMoE is not random but follows a **systematic depth-dependent progression**. This specialization is directly enabled by the global token pool, which provides each expert with a diverse and representative sample of the current batch's computational demands, allowing them to learn distinct temporal roles.
>
> The visualization of expert capacity utilization across timesteps and layers `(Appendix Figure 9 in our paper)` reveals this evolutionary trajectory:
>
> *   **Layer 1 (shallow)**: Experts act as dynamic "generalists," exhibiting the **"Bookends"** pattern with high capacity utilization during both high-noise (early) and low-noise (late) steps. This indicates their dual role in initial noisy feature parsing and final detail refinement.
>
> *   **Layer 7 (intermediate)**: Routing becomes more stable and balanced. Experts here predominantly follow the **"Mid-Peak"** pattern, concentrating computation on the semantically complex mid-range timesteps
> *   **Layer 13 (deep-mid)** shows smooth, long-range capacity transitions. Experts here handle fewer early-step tokens and focus more on structured representations emerging in middle and later timesteps.
> *   **Layer 19 (deep)**: Deep-layer experts demonstrate highly stable and specialized behavior, primarily as **"Anchors"** with consistent engagement. They focus on processing structurally coherent representations during the critical middle-to-late stages, specializing in high-level semantic refinement.
>
> From macro-view of all experts, we also observe a consistent global trend: **experts tend to receive more tokens in middle timesteps**, aligning with our Figure 1(b) and findings reported in SD3 [1] regarding the difficulty of mid-range timesteps for velocity prediction.
>
> These observations confirm that **experts do not operate uniformly but gradually specialize**, enabled by the batch-level global token pool and capacity predictor. By exposing experts to a diverse and globally aggregated set of tokens at each step, the global pool encourages specialization toward different noise levels and semantic complexities.
>
> #### **A Taxonomy of Expert Temporal Specialization**
>
> We further roughly categorize the observed expert behaviors into five distinct temporal specialization patterns, which are systematically distributed across the network depth:
>
> 1.  **The Bookends:** Highly activated at the start and end of the denoising process.
> 2.  **The Mid-Peak:** Most active during the middle stages.
> 3.  **The Crescendo:** Activity gradually increases over time.
> 4.  **The Diminuendo:** Activity gradually decreases over time.
> 5.  **The Anchor:** Maintains a consistent level of activity throughout all steps.
>
> `Each sub figures have been tagged by the categories above in revised paper.`
>
> #### **Summary of Expert Behavior by Layer**
>
> | Network Layer | Predominant Routing Pattern(s) | Inferred Specialization |
> | :- | :-| :--|
> | **Layer 1 (Shallow)** | **The Bookends (Primary)**, The Mid-Peak | Low-level, noise-robust feature extraction|
> | **Layer 7 (Mid)** | **The Mid-Peak (Primary)**, The Crescendo, The Diminuendo | Mid-level feature integration and structuring  |
> | **Layer 13 (Deep-Mid)** | **The Mid-Peak (Primary)**, The Crescendo, The Anchor | Refined structural and conceptual representation |
> | **Layer 19 (Deep)** | The Mid-Peak, The Crescendo, **The Anchor (Primary)**| High-level semantic refinement and detail polishing |
>
> ---

---

> > ### Author Response · Authors · 2025-11-20
> >
> > ### **2. Macro-Computation Allocation Analysis**
> >
> > In **Appendix Section B (Dynamic Conditional Computation)**, we extend this analysis to the macro scale. We demonstrate that DiffMoE's specialization mechanism enables **task-adaptive computation allocation**.
> >
> > **Takeaway:** The global token pool is the cornerstone of this architecture. By providing a global view of the batch's computational needs at each denoising step, it naturally encourages experts to **diversify and specialize** across the noise spectrum and network hierarchy, forming an efficient, adaptive computational network. DiffMoE dynamically adjusts its total expert utilization and routing distribution between C2I (Class-to-Image) and T2I (Text-to-Image) tasks. This shows that the model not only specializes experts internally but also leverages this specialization to reallocate computational resources at a system level based on timesteps and conditions.
> >
> > *Reference*
> > [1] *Scaling Rectified Flow Transformers for High-Resolution Image Synthesis*

---

> > > ### Author Response · Authors · 2025-11-20
> > >
> > > **Q2: About generalization of capacity predictor**
> > >
> > > > Q2:How well does the capacity predictor generalize across different datasets or tasks? Does it require retraining for each new domain?
> > >
> > > **[Reply]**
> > >
> > > Thank you for the insightful question.
> > >
> > > If we shift the model to a **fundamentally different task**, such as text-to-depth (modality change) or text-to-video (temporal consistency learning), **retraining is required**. This is because the capacity predictor is an integral component of the model, and it must be fine-tuned to adapt to the new task-specific routing patterns.
> > >
> > > However, if the **task remains the same**—for example, staying within text-to-image but switching to a different dataset—**retraining is not necessary**. The capacity predictor generalizes robustly across datasets within the same task domain.
> > >
> > > - **Reason:** It learns the underlying principles of token routing during initial training on large, diverse datasets, rather than memorizing dataset-specific content.
> > > - **Evidence:** This enables strong zero-shot transfer to novel prompts and unseen concepts, leveraging the inherent generalization ability of the base diffusion model.
> > >
> > >
> > >
> > > **W2/Q3: About high-resolution tasks and scalability**
> > >
> > > > W2:In addition, the batch-level global token pool may pose scalability challenges when extending to high-resolution images or video generation.
> > > >
> > > > Q3:How does DiffMoE perform on high-resolution tasks, and are there any scalability limitations?
> > >
> > > **[Reply]**
> > >
> > > ### **1. Scalability on High-Resolution Image Generation**
> > >
> > > Thank you for the insightful comments. To evaluate whether the batch-level global token pool limits scalability, we conducted additional experiments on ImageNet at high resolution using **8 GPUs** with a **local batch size (token-pool size) of only 8**. This setup intentionally stresses the token-pool design under constrained batch conditions.
> > >
> > > As shown below, **DiffMoE continues to outperform the dense baseline**, even when the global pool is smaller:
> > >
> > > | Model (3000K) | Active Params (M) | FID50K↓ (CFG=1.5) |
> > > | ------------- | ----------------- | ----------------- |
> > > | Dense-L       | 458M              | 5.8               |
> > > | DiffMoE-L     | 458M              | **4.0**           |
> > >
> > > These results indicate that **DiffMoE scales well to high-resolution image generation** and does not exhibit noticeable degradation due to limited batch size or reduced pool diversity.
> > >
> > > ------
> > >
> > > ### **2. Toward Video Generation & Ultra-High-Resolution Models**
> > >
> > > For video or ultra-high-resolution generation, the effective batch size per GPU may become even smaller, potentially reducing the diversity of tokens in the global pool. We discuss three scalable implementations that ensure efficient routing under such conditions:
> > >
> > > ------
> > >
> > > #### **(1) Full Synchronization (Accurate but Expensive)**
> > >
> > > All gating scores **[B, S, E]** are gathered across GPUs to construct a true global token pool. A global top-k is computed and redistributed back to each rank.  ( *B: local batch size, S: sequence length, E: number of experts* )
> > >
> > > - **Pros:** Routing is exact.
> > > - **Cons:** Communication overhead grows with resolution and number of experts.
> > >
> > > ------
> > >
> > > #### **(2) Group-Wise Pooling (Intra-Node Approximation)**
> > >
> > > GPUs within the same node form a local group to build a shared token pool using high-bandwidth intra-node communication.
> > >
> > > - **Pros:** Much lower communication cost; still preserves cross-sample diversity.
> > > - **Cons:** Slight approximation to the true global distribution, but sufficient with a large enough pool.
> > >
> > > ------
> > >
> > > #### **(3) Quantile-Based Approximation (Highly Efficient)**
> > >
> > > Instead of syncing full gating scores,  each GPU computes local K-quantiles of gating scores. These quantiles ($N_{Node}\times N_{GPUs\ per\ Node}$, which are much fewer in number than raw token scores) can be aggregated across GPUs with minimal communication.  These small quantile tensors are aggregated globally and averaged to estimate a **global routing threshold**.
> > >
> > > - **Pros:** Communication cost becomes **nearly constant**, independent of sequence length.
> > > - **Cons:** Approximate threshold, but works well in practice for robust routing.
> > >
> > > ------
> > >
> > > ### **Summary**
> > >
> > > These strategies provide a flexible spectrum from exact synchronization to lightweight approximations. Selecting the appropriate mode allows **DiffMoE to scale effectively to video and ultra-high-resolution generation**, even when the per-GPU batch size is small. We consider this an exciting direction for further exploration in future work.

---

### Official Review · Reviewer_YtiD · 2025-10-30

**Soundness:** 3
**Presentation:** 3
**Contribution:** 3
**Rating:** 6
**Confidence:** 5

**Summary:**

This paper introduces DiffMoE, a novel Mixture-of-Experts (MoE) architecture designed for diffusion transformers. By leveraging a batch-level global token pool, a capacity predictor, and a dynamic threshold, DiffMoE enables efficient token selection and adaptive computation across varying noise levels and conditions. It achieves state-of-the-art performance in both class-conditional and text-to-image generation tasks, surpassing dense models with fewer activated parameters. DiffMoE demonstrates superior scaling efficiency and broader applicability, addressing limitations in prior MoE-based diffusion models.

**Strengths:**

+ The proposed DiffMoE is useful for  building scalebale diffusion model.
+ The proposed dynamic computation is interesting and reasonable

**Weaknesses:**

- The proposed method, DiffMoE, should be clearly highlighted in each table to enhance readability.

- Since the key advantage of MoE lies in its scalability, it would be beneficial to train models of varying sizes to further explore this aspect.

- Lack of a comprehensive literature review. For example, the use of MoE for diffusion transformers [1] and diffusion models with dynamic computation [2] should be discussed in the related work section.



[1] Diff-MoE: Diffusion Transformer with Time-Aware and Space-Adaptive Experts, ICML 2025.

[2] Dynamic diffusion transformer, ICLR 2025

**Questions:**

see weakness

---

> ### Author Response · Authors · 2025-11-20
>
> We sincerely thank the reviewer for the positive comments on our work! We address the questions and clarify the issues accordingly as described below.
>
> > W denotes Weakness, Q denotes Question.
>
> **W1: About highlighting our method**
>
> > W1: The proposed method, DiffMoE, should be clearly highlighted in each table to enhance readability.
>
> **[Reply]**
>
> Thank you for the helpful suggestion. We have revised all tables in the updated version and **clearly highlighted our method (DiffMoE)** to improve readability and make comparisons more intuitive.
>
>
>
> **W2: About varying size of our model**
>
> > W2: Since the key advantage of MoE lies in its scalability, it would be beneficial to train models of varying sizes to further explore this aspect.
>
> **[Reply]**
>
> Thank you for your valuable suggestions. As shown in Figure 4 of the submitted paper, we provide DiffMoE models across multiple scales from S to L. Table 2 further illustrates the strong parameter-scaling behavior of DiffMoE: for example, our 458M-active-parameter DiffMoE already surpasses its dense counterpart with 1353M active parameters. **Overall, DiffMoE supports a broad spectrum of active model sizes—33M, 130M, 458M, and 675M—and consistently exhibits strong performance on both C2I and large-scale T2I tasks.** Additional quantitative comparisons are provided in the table below:
>
> | Model Size | Method / 400K / Flow |           |
> | ---------- | -------------------- | --------- |
> | S/33M      | TC-DiT-S-E16         | 56.76     |
> | S/33M      | Dense-DiT-S          | 59.84     |
> | S/33M      | EC-DiT-S-E16         | 57.79     |
> | S/33M      | **DiffMoE-S-E16**    | **54.61** |
> | B/130M     | TC-DiT-B-E16         | 37.96     |
> | B/130M     | Dense-DiT-B          | 36.24     |
> | B/130M     | EC-DiT-B-E16         | 34.64     |
> | B/130M     | **DiffMoE-B-E16**    | **32.89** |
> | L/458M     | TC-DiT-L-E16         | 24.60     |
> | L/458M     | Dense-DiT-L          | 21.16     |
> | L/458M     | EC-DiT-L-E16         | 21.76     |
> | L/458M     | **DiffMoE-L-E16**    | **18.91** |
> | XL/675M    | Dense-DiT-XL         | 18.80     |
> | XL/675M    | **DiffMoE-XL-E16**   | **17.63** |
>
> The aforementioned results are based on the Flow Matching method. To complement this, we also present a DDPM-trained version contained S, B, L sizes in Appendix Figure 6. The consistent high performance across both training paradigms robustly demonstrates the general effectiveness of our method.
>
>
>
> **W3: More Discussion on the Two Cited Works**
>
> > W3: Lack of a comprehensive literature review. For example, the use of MoE for diffusion transformers [1] and diffusion models with dynamic computation [2] should be discussed in the related work section.
>
> **[Reply]**
>
> Thank you for raising this important point. We have expanded the discussion of related work in the revised paper (using blue font). More detailed analysis is shown below:
>
> Diff-MoE [1] proposes a MoE architecture with **time-aware and space-adaptive routing**, which is fundamentally orthogonal to our approach. Their method explicitly adapts expert activation based on diffusion timesteps and spatial context. In contrast, DiffMoE focuses on **global batch-level token routing and capacity prediction**, allowing the model to implicitly learn routing behavior **without introducing additional hand-crafted inductive biases**. These two directions are complementary, and integrating them could potentially lead to a more powerful and flexible MoE-based diffusion model.
>
> DyDiT [2] introduces a dynamic computation mechanism designed primarily to **reduce inference-time cost** for diffusion transformers. This direction is orthogonal to ours: DyDiT focuses on **runtime efficiency**, while DiffMoE targets **architectural design and its scalability** during training and modeling. In other words, DyDiT adapts computation per timestep at inference, whereas DiffMoE restructures the model itself through MoE-based expert routing and capacity allocation. These approaches are complementary, and combining DiffMoE with dynamic-inference techniques such as DyDiT offers a promising avenue for future work.
>
>
>
> *Reference*
>
> *[1] Diff-MoE: Diffusion Transformer with Time-Aware and Space-Adaptive Experts, ICML 2025.*
>
> *[2] Dynamic Diffusion Transformer, ICLR 2025.*

---

> > ### Comment · Reviewer_YtiD · 2025-11-28
> >
> > Thank you for your response. My concerns have been well addressed, and I hope this paper will be accepted to ICLR.
> >
> > However, I suggest avoiding the use of too many different colors in the paper, as it may make it appear overly flashy. I kindly ask the authors to consider this.

---

> ### Author Response · Authors · 2025-11-29
>
> Dear Reviewer YtiD,
>
> Thank you very much for your thoughtful feedback and the strong support for our paper. We are delighted to hear that your concerns have been well addressed.
>
> We sincerely appreciate your suggestion regarding the use of colors. **We have revised paper to reduce color use.** To ensure clear distinction without being overly flashy, we now emphasize the **best results in bold** and the **second-best results with an underline.**
>
> Thank you once again for your constructive comments and positive recommendation.
>
> Sincerely,
>
> The Authors

---

### Author Response · Authors · 2025-11-29
**Subject: General Response to Area Chair and Reviewers: Summary of Revisions and Additional Experiments**

Dear Area Chair and Reviewers,

We sincerely appreciate the time and effort you have dedicated to reviewing our manuscript.

We introduce **DiffMoE**, a novel Mixture-of-Experts (MoE) architecture that enables efficient and adaptive computation in Diffusion Transformers (DiTs) by using a **Batch-Level Global Token Pool**,  a **Capacity Predictor** and **Dynamic Threshold** technique. As the reviewers highlighted, the paper presents a promising and well-motivated method (jjxW), introduces useful, interesting, and reasonable design choices (YtiD), employs a fair comparison framework (jjxW), and is supported by strong empirical results demonstrating state-of-the-art performance with significantly fewer activated parameters (all).

We greatly appreciate your constructive feedback. In response, we have carefully revised and enhanced the manuscript by providing extensive additional analysis and experiments:

- **In-depth Analysis of Expert Specialization** (Appendix B.2, Response to bR6L): We provided a comprehensive analysis, identifying systematic expert specialization patterns (e.g., *Bookends, Mid-Peak, Anchor*) enabled by the Global Token Pool.
- **Robustness and Scalability Studies** (Response to jjxW, bR6L): We demonstrated DiffMoE's robustness under small-batch training/inference and discussed its scalability to high-resolution and video generation tasks.
- **Expanded Qualitative and Quantitative Comparisons**: We added more visual comparisons (Fig. 13, Response to jjxW) for Text-to-Image generation and provided additional quantitative results (Response to YtiD) across various model scales.
- **Enhanced Presentation:** Following Reviewer YtiD's suggestion, we have **revised all tables to reduce excessive coloring**. We now use a clear formatting scheme (**Bold** for best, **Underline** for second-best) for emphasis.
- **Expanded Related Work:** (Response to YtiD) We have clarified the relationship between DiffMoE and orthogonal dynamic diffusion models (*DyDiT*) and other MoE approaches (*Diff-MoE*).

In the revised manuscript, these updates are temporarily highlighted in **blue** for your convenience.

We sincerely hope these updates help better convey the benefits of the proposed DiffMoE to the ICLR community. We look forward to further discussion.

Thank you very much,

Authors.

---

### Meta-Review · Area_Chair_d3op · 2026-01-03

**Summary:**

This submission presents a method for scalable and efficient MoE-based diffusion models. The key finding in this work is that access to the global token distribution can help to apply MoE to diffusion models. A model can learn and dynamically process the tokens from different noise levels and conditions. This allows experts to be trained simultaneously on a diverse set of tokens from multiple images and at various noise levels. To optimize token selection during inference, the submission proposes a capacity predictor that dynamically adjusts resource allocation (efficiently distributing computational resources between complex and simple cases).

The work demonstrates that DiffMoE improves over dense baselines with fewer activated parameters (3x) and slightly outperforms existing MoE approaches for diffusion models. The submission is a step forward in solving the problem that existing attempts to integrate MoE with diffusion models have yielded suboptimal results, namely failing to achieve the remarkable improvements observed in language models.

I would expect the reviewers to point out that this paper introduces related work at the beginning of Sections 2 and 4. This is an obvious redundancy. These two subsections on diffusion models and the MoE should be merged into one Section 2: "Related Work and Bacground". The gained space should be used to add experiments suggested by reviewers, especially from the reviewer bR6L. Overall, the paper should be extended to 10 pages. I am really disappointed with this part of this paper. What is the 3rd key dimension at the beginning of Section 3? At the end of section 3.2, it should be MoE and not moe. For the "Results are shown" an incorrect reference is used and no Table is shown.  The results in Table 3 for the T2I COCO dataset are not indicating a "remarkable improvements" for MoE with diffusion models (as compared to EC-DiT). The difference between Dense-DiT and EC-DiT is significantly larger (e.g., FID10k drop by 1.58) than the difference between EC-DiT and DiffMoE, which is only 0.33.

This paper received only 3 reviews. I would like to have more input from the reviewers (at least one more review) and an answer from the reviewer bR6L to the author's rebuttal. I agree with the reviewer bR6L comments and, for example, would like to see a clear demonstration of how DiffMoE method generalizes between different tasks and datasets. One of the experiments that the authors should add is to quantify (clearly demonstrate experimentally) how the capacity predictor generalizes robustly across datasets within the same task domain.

While I appreciate the authors' efforts in addressing an important problem, I regret that I cannot recommend acceptance of this paper in its current form. The primary concern is that the experimental evidence does not sufficiently demonstrate that the proposed method offers substantial improvements over the baselines like the EC-DiT approach. Additionally, the paper would benefit considerably from significant revisions to its presentation, organization, and experimental validation. I would like to emphasize that these concerns stem from the paper's current execution rather than the underlying research direction, which I believe is very important. I encourage the authors to address the above issues and consider resubmitting their work after making these improvements. I believe that with the necessary revisions, this research could reach the quality standards expected for publication.

**Reviewer Concerns:**

I agree with the reviewer bR6L comments and, for example, would like to see a clear demonstration of how DiffMoE method generalizes between different tasks and datasets. One of the experiments that the authors should add is to quantify (clearly demonstrate experimentally) how the capacity predictor generalizes robustly across datasets within the same task domain.

Reviewer YtiD requested training models of varying sizes to further explore the scalability aspect and it was provided.

Reviewer jjxW requested the additional ablation studies on the batch sizes that were executed.

**Reviewer Scores:**

YtiD: 6

qLWV: Review not provided

bR6L: 4

jjxW: 6

---

### Decision · Program_Chairs · 2026-01-26

Reject